# Optimal Sketching for Trace Estimation

**Shuli Jiang**
Robotics Institute
Carnegie Mellon University
shulij@andrew.cmu.edu

**Hai Pham**
Language Technologies Institute
Carnegie Mellon University
htpham@cs.cmu.edu

**David P. Woodruff**
Computer Science Department
Carnegie Mellon University
dwoodruf@cs.cmu.edu

**Qiuyi (Richard) Zhang**
Google Brain
qiuyiz@google.com

## Abstract

Matrix trace estimation is ubiquitous in machine learning applications and has traditionally relied on Hutchinson's method, which requires $O(\log(1/\delta)/\epsilon^2)$ matrix-vector product queries to achieve a $(1 \pm \epsilon)$-multiplicative approximation to $\text{tr}(A)$ with failure probability $\delta$ on positive-semidefinite input matrices $A$. Recently, the Hutch++ algorithm was proposed, which reduces the number of matrix-vector queries from $O(1/\epsilon^2)$ to the optimal $O(1/\epsilon)$, and the algorithm succeeds with constant probability. However, in the high probability setting, the non-adaptive Hutch++ algorithm suffers an extra $O(\sqrt{\log(1/\delta)})$ multiplicative factor in its query complexity. Non-adaptive methods are important, as they correspond to sketching algorithms, which are mergeable, highly parallelizable, and provide low-memory streaming algorithms as well as low-communication distributed protocols. In this work, we close the gap between non-adaptive and adaptive algorithms, showing that even non-adaptive algorithms can achieve $O(\sqrt{\log(1/\delta)}/\epsilon + \log(1/\delta))$ matrix-vector products. In addition, we prove matching lower bounds demonstrating that, up to a $\log \log(1/\delta)$ factor, no further improvement in the dependence on $\delta$ or $\epsilon$ is possible by any non-adaptive algorithm. Finally, our experiments demonstrate the superior performance of our sketch over the adaptive Hutch++ algorithm, which is less parallelizable, as well as over the non-adaptive Hutchinson's method.

## 1   Introduction

The problem of implicit matrix trace estimation arises naturally in a wide range of applications [1]. For example, during the training of Gaussian Process, a popular non-parametric kernel-based method, the calculation of the marginal log-likelihood contains a heavy-computation term, i.e., the log determinant of the covariance matrix, $\log(\det(\mathbf{K}))$, where $\mathbf{K} \in \mathbb{R}^{n \times n}$, and $n$ is the number of data points. The canonical way of computing $\log(\det(\mathbf{K}))$ is via Cholesky decomposition on $\mathbf{K}$, whose time complexity is $O(n^3)$. Since $\log(\det(\mathbf{K})) = \sum_{i=1}^{n} \log(\lambda_i)$, where $\lambda_i$'s are the eigenvalues of $\mathbf{K}$, one can compute $\text{tr}(\log(\mathbf{K}))$ instead. Trace estimation combined with polynomial approximation (e.g., the Chebyshev polynomial or Stochastic Lanczos Quadrature) to $\log$ [2], or trace estimation combined with maximum entropy estimation [3] provide fast ways of estimating $\text{tr}(\log(\mathbf{K}))$ for large-scale data. Other popular applications of implicit trace estimation include counting triangles and computing the Estrada Index in graphs [4, 5], approximating the generalized rank of a matrix [6], and studying non-convex loss landscapes from the Hessian matrix of large neural networks (NNs) [7, 8].

To define the problem, we consider the *matrix-vector product model* as formalized in [9, 10], where there is a real symmetric input matrix $\boldsymbol{A} \in \mathbb{R}^{n \times n}$ that cannot be explicitly presented but one has oracle access to $\boldsymbol{A}$ via matrix-vector queries, i.e., one can obtain $\boldsymbol{A}\mathbf{q}$ for any desired query vector $\mathbf{q} \in \mathbb{R}^n$. For example, due to a tremendous amount of trainable parameters of large NNs, it is often prohibitive to compute or store the entire Hessian matrix $\boldsymbol{H}$ with respect to some loss function from the parameters [7], which is often used to study the non-convex loss landscape. However, with Pearlmutter's trick [11] one can compute $\boldsymbol{H}\mathbf{q}$ for any chosen vector $\mathbf{q}$. The goal is to efficiently estimate the trace of $\boldsymbol{A}$, denoted by $\mathrm{tr}(\boldsymbol{A})$, up to $\epsilon$ error, i.e., to compute a quantity within $(1 \pm \epsilon)\mathrm{tr}(\boldsymbol{A})$. For efficiency, such algorithms are randomized and succeed with probability at least $1 - \delta$. The minimum number of queries $q$ required to solve the problem is referred to as the *query complexity*.

Computing matrix-vector products $\boldsymbol{A}\mathbf{q}$ through oracle access, however, can be costly. For example, computing Hessian-vector products $\boldsymbol{H}\mathbf{q}$ on large NNs takes approximately twice the time of backpropagation. When estimating the eigendensity of $\boldsymbol{H}$, one computes $\mathrm{tr}(f(\boldsymbol{H}))$ for some density function $f$, and needs repeated access to the matrix-vector product oracle. As a result, even with Pearlmutter's trick and distributed computation on modern GPUs, it takes 20 hours to compute the eigendensity of a single Hessian $\boldsymbol{H}$ with respect to the cross-entropy loss on the CIFAR-10 dataset [12], from a set of fixed weights for ResNet-18 [13] which has approximately 11 million parameters [7]. Thus, it is important to understand the fundamental limits of implicit trace estimation as the query complexity in terms of the desired approximation error $\epsilon$ and the failure probability $\delta$.

Hutchinson's method [14], a simple yet elegant randomized algorithm, is the ubiquitous work force for implicit trace estimation. Letting $\boldsymbol{Q} = [\mathbf{q}_1, \ldots, \mathbf{q}_q] \in \mathbb{R}^{n \times q}$ be $q$ vectors with i.i.d. Gaussian or Rademacher (i.e., $\pm 1$ with equal probability) random variables, Hutchinson's method returns an estimate of $\mathrm{tr}(\boldsymbol{A})$ as $\frac{1}{q} \sum_{i=1}^{q} \mathbf{q}_i^T \boldsymbol{A} \mathbf{q}_i = \frac{1}{q}\mathrm{tr}(\boldsymbol{Q}^T \boldsymbol{A} \boldsymbol{Q})$. Although Hutchinson's method dates back to 1990, it is surprisingly not well-understood on positive semi-definite (PSD) matrices. It was originally shown that for PSD matrices $\boldsymbol{A}$ with the $\mathbf{q}_i$ being Gaussian random variables, in order to obtain a multiplicative $(1 \pm \epsilon)$ approximation to $\mathrm{tr}(\boldsymbol{A})$ with probability at least $1 - \delta$, $O(\log(1/\delta)/\epsilon^2)$ matrix-vector queries suffice [15].

A recent work [16] proposes a variance-reduced version of Hutchinson's method that shows only $O(1/\epsilon)$ matrix-vector queries are needed to achieve a $(1 \pm \epsilon)$-approximation to any PSD matrix with constant success probability, in contrast to the $O(1/\epsilon^2)$ matrix-vector queries needed for Hutchinson's original method. The key observation is that the variance of the estimated trace in Hutchinson's method is largest when there is a large gap between the top few eigenvalues and the remaining ones. Thus, by splitting the number of matrix-vector queries between approximating the top $O(1/\epsilon)$ eigenvalues, i.e., by computing a rank-$O(1/\epsilon)$ approximation to $\boldsymbol{A}$, and performing trace estimation on the remaining part of the spectrum, one needs only $O(1/\epsilon)$ queries in total to achieve a $(1 \pm \epsilon)$ approximation to $\mathrm{tr}(\boldsymbol{A})$. Furthermore, [16] shows $\Omega(1/\epsilon)$ queries are in fact necessary for *any* trace estimation algorithm, up to a logarithmic factor, for algorithms succeeding with constant success probability. While [16] mainly focuses on the improvement on $\epsilon$ in the query complexity with constant failure probability, we focus on the dependence on the failure probability $\delta$.

---

**Algorithm 1** `Hutch++`: Stochastic trace estimation with **adaptive** matrix-vector queries

---
1: **Input:** Matrix-vector multiplication oracle for PSD matrix $\boldsymbol{A} \in \mathbb{R}^{n \times n}$. Number $m$ of queries.
2: **Output:** Approximation to $\mathrm{tr}(\boldsymbol{A})$.
3: Sample $\boldsymbol{S} \in \mathbb{R}^{n \times \frac{m}{3}}$ and $\boldsymbol{G} \in \mathbb{R}^{n \times \frac{m}{3}}$ with i.i.d. $\mathcal{N}(0, 1)$ entries.
4: Compute an orthonormal basis $\boldsymbol{Q} \in \mathbb{R}^{n \times \frac{m}{3}}$ for the span of $\boldsymbol{AS}$ via $\boldsymbol{QR}$ decomposition.
5: **return** $t = \mathrm{tr}(\boldsymbol{Q}^T \boldsymbol{A} \boldsymbol{Q}) + \frac{3}{m}\mathrm{tr}(\boldsymbol{G}^T(\boldsymbol{I} - \boldsymbol{QQ}^T)\boldsymbol{A}(\boldsymbol{I} - \boldsymbol{QQ}^T)\boldsymbol{G})$.

---

**Algorithm 2** `NA-Hutch++`: Stochastic trace estimation with **non-adaptive** matrix-vector queries

---
1: **Input:** Matrix-vector multiplication oracle for PSD matrix $\boldsymbol{A} \in \mathbb{R}^{n \times n}$. Number $m$ of queries.
2: **Output:** Approximation to $\mathrm{tr}(\boldsymbol{A})$.
3: Fix constants $c_1, c_2, c_3$ such that $c_1 < c_2$ and $c_1 + c_2 + c_3 = 1$.
4: Sample $\boldsymbol{S} \in \mathbb{R}^{n \times c_1 m}$, $\boldsymbol{R} \in \mathbb{R}^{n \times c_2 m}$, and $\boldsymbol{G} \in \mathbb{R}^{n \times c_3 m}$, with i.i.d. $\mathcal{N}(0, 1)$ entries.
5: $\boldsymbol{Z} = \boldsymbol{AR}, \boldsymbol{W} = \boldsymbol{AS}$
6: **return** $t = \mathrm{tr}((\boldsymbol{S}^T \boldsymbol{Z})^\dagger(\boldsymbol{W}^T \boldsymbol{Z})) + \frac{1}{c_3 m}(\mathrm{tr}(\boldsymbol{G}^T \boldsymbol{A} \boldsymbol{G}) - \mathrm{tr}(\boldsymbol{G}^T \boldsymbol{Z}(\boldsymbol{S}^T \boldsymbol{Z})^\dagger \boldsymbol{W}^T \boldsymbol{G}))$.

---

Achieving a low failure probability $\delta$ is important in applications where failures are highly undesirable, and the low failure probability regime is well-studied in related areas such as compressed sensing [17], data stream algorithms [18, 19], distribution testing [20], and so on. While one can always reduce the failure probability from a constant to $\delta$ by performing $O(\log(1/\delta))$ independent repetitions

and taking the median, this multiplicative overhead of $O(\log(1/\delta))$ can cause a huge slowdown in practice, e.g., in the examples above involving large Hessians.

Two algorithms were proposed in [16]: `Hutch++` (**Algorithm** 1), which requires *adaptively* chosen matrix-vector queries and `NA-Hutch++` (**Algorithm** 2) which only requires *non-adaptively* chosen queries. We call the matrix-vector queries adaptively chosen if subsequent queries are dependent on previous queries $\mathbf{q}$ and observations $\mathbf{Aq}$, whereas the algorithm is non-adaptive if all queries can be chosen at once without any prior information about $\mathbf{A}$. Note that Hutchinson's method uses only non-adaptive queries. [16] shows that `Hutch++` can use $O(\sqrt{\log(1/\delta)}/\epsilon + \log(1/\delta))$ adaptive matrix-vector queries to achieve $(1 \pm \epsilon)$ approximation with probability at least $1 - \delta$, while `NA-Hutch++` can use $O(\log(1/\delta)/\epsilon)$ non-adaptive queries. Thus, in many parameter regimes the non-adaptive algorithm suffers an extra $\sqrt{\log(1/\delta)}$ multiplicative factor over the adaptive algorithm.

It is important to understand the query complexity of non-adaptive algorithms for trace estimation because the advantages of non-adaptivity are plentiful: algorithms that require only non-adaptive queries can be easily parallelized across multiple machines while algorithms with adaptive queries are inherently sequential. Furthermore, non-adaptive algorithms correspond to sketching algorithms which are the basis for many streaming algorithms with low memory [21] or distributed protocols with low-communication overhead (for an example application to low rank approximation, see [22]). We note that there are numerous works on estimating matrix norms in a data stream [23, 24, 25, 26], most of which use trace estimation as a subroutine.

## 1.1 Our Contributions

**Improving the Non-adaptive Query Complexity.** We give an improved analysis of the query complexity of the non-adaptive trace estimation algorithm `NA-Hutch++` (**Algorithm** 2), based on a new low-rank approximation algorithm and analysis in the high probability regime, instead of applying an off-the-shelf low-rank approximation algorithm as in [16]. Instead of $O(\log(1/\delta)/\epsilon)$ queries as shown in [16], we show that $O(\sqrt{\log(1/\delta)}/\epsilon + \log(1/\delta))$ non-adaptive queries suffice to achieve a multiplicative $(1 \pm \epsilon)$ approximation of the trace with probability at least $1 - \delta$, which matches the query complexity of the adaptive trace estimation algorithm `Hutch++`. Since our algorithm is non-adaptive, it can be used in subroutines in streaming and distributed settings for estimating the trace, with lower memory than was previously possible for the same failure probability.

**Theorem 1.1** (Restatement of Theorem 3.1). *Let $\mathbf{A}$ be any PSD matrix. If `NA-Hutch++` is implemented with $m = O\left(\frac{\sqrt{\log(1/\delta)}}{\epsilon} + \log(1/\delta)\right)$ matrix-vector multiplication queries, then with probability $1 - \delta$, the output $t$ of `NA-Hutch++` satisfies $(1 - \epsilon)\mathrm{tr}(\mathbf{A}) \leq t \leq (1 + \epsilon)\mathrm{tr}(\mathbf{A})$.*

The improved dependence on $\delta$ is perhaps surprising in the non-adaptive setting, as simply repeating a constant-probability algorithm would give an $O(\log(1/\delta)/\epsilon)$ dependence. Our non-adaptive algorithm is as good as the best known adaptive algorithm, and much better than previous non-adaptive algorithms [16, 14]. The key difference between our analysis and the analysis in [16] is in the number of non-adaptive matrix-vector queries we need to obtain an $O(1)$-approximate rank-$k$ approximation to $\mathbf{A}$ in Frobenius norm.

Specifically, to reduce the total number of matrix-vector queries, our queries are split between (1) computing $\tilde{\mathbf{A}}$, a rank-$k$ approximation to the matrix $\mathbf{A}$, and (2) performing trace estimation on $\mathbf{A} - \tilde{\mathbf{A}}$. Let $\mathbf{A}_k = \min_{\text{rank-}k \ \mathbf{A}} \|\mathbf{A} - \mathbf{A}_k\|_F$ be the best rank-$k$ approximation to $\mathbf{A}$ in Frobenius norm. For our algorithm to work, we require $\|\mathbf{A} - \tilde{\mathbf{A}}\| \leq O(1)\|\mathbf{A} - \mathbf{A}_k\|_F$ with probability $1 - \delta$. Previous results from [27] show the number of non-adaptive queries required to compute $\tilde{\mathbf{A}}$ is $O(k \log(1/\delta))$, where each query is an i.i.d. Gaussian or Rademacher vector. We prove $O(k + \log(1/\delta))$ non-adaptive Gaussian query vectors suffice to compute $\tilde{\mathbf{A}}$. Low rank approximation requires both a so-called subspace embedding and an approximate matrix product guarantee (see, e.g., [28], for a survey on sketching for low rank approximation), and we show both hold with the desired probability, with some case analysis, for Gaussian queries. A technical overview can be found in Section 3.

The improvement on the number of non-adaptive queries to achieve $O(1)$-approximate rank-$k$ approximation has many other implications, which can be of an independent interest. For example, since low-rank approximation algorithms are extensively used in streaming algorithms suitable for low-memory settings, this new result directly improves the space complexity of the state-of-the-art

streaming algorithm for Principle Component Analysis (PCA) [29] from $O(d \cdot (k \log(1/\delta)))$ to $O(d \cdot (k + \log(1/\delta)))$ for constant approximation error $\epsilon$, where $d$ is the dimension of the input.

**Lower Bound.** Previously, no lower bounds were known on the query complexity in terms of $\delta$ in a high probability setting. In this work, we give a novel matching lower bound for non-adaptive (i.e., sketching) algorithms for trace estimation, with novel techniques based on a new family of hard input distributions, showing that our improved $O(\sqrt{\log(1/\delta)}/\epsilon + \log(1/\delta))$ upper bound is optimal, up to a $\log \log(1/\delta)$ factor, for any $\epsilon \in (0, 1)$. The methods previously used to prove an $\Omega(1/\epsilon)$ lower bound with constant success probability (up to logarithmic factors) in [16] do not apply in the high probability setting. Indeed, [16] gives two lower bound methods based on a reduction from two types of problems: (1) a communication complexity problem, and (2) a distribution testing problem between clean and negatively spiked random covariance matrices. Technique (1) does not apply since there is not a multi-round lower bound for the Gap-Hamming communication problem used in [16] that depends on $\delta$. One might think that since we are proving a non-adaptive lower bound, we could use a non-adaptive lower bound for Gap-Hamming (which exists, see [18]), but this is wrong because even the non-adaptive lower bound in [16] uses a 2-round lower bound for Gap-Hamming, and there is no such lower bound known in terms of $\delta$. Technique (2) also does not apply, as it involves a $1/\epsilon \times 1/\epsilon$ matrix, which can be recovered exactly with $1/\epsilon$ queries; further, increasing the matrix dimensions would break the lower bound as their two cases would no longer need to be distinguished. Thus, such a hard input distribution fails to show the additive $\Omega(\log(1/\delta))$ term in the lower bound.

Our starting point for a hard instance is a family of Wigner matrices (see Definition 2.1) shifted by an identity matrix so that they are PSD. However, due to strong concentration properties of these matrices, they can only be used to provide a lower bound of $\Omega(\sqrt{\log(1/\delta)}/\epsilon)$ when $\epsilon < 1/\sqrt{\log(1/\delta)}$. Indeed, setting $\delta$ to be a constant in this case recovers the $\Omega(1/\epsilon)$ lower bound shown in [16] but via a completely different technique. For larger $\epsilon$, we consider a new distribution testing problem between clean Wigner matrices and the same distribution with a large rank-1 noisy PSD matrix, and then argue with probability roughly $\delta$, all non-adaptive queries have unusually tiny correlation with this rank-1 matrix, thus making it indistinguishable between the two distributions. This gives the desired additive $\Omega(\log(1/\delta))$ lower bound, up to a $\log \log(1/\delta)$ factor.

**Theorem 1.2** (Restatement of Theorem 4.1). *Suppose $\mathcal{A}$ is a non-adaptive query-based algorithm that returns a $(1 \pm \epsilon)$-multiplicative estimate to $\mathrm{tr}(\boldsymbol{A})$ for any PSD matrix $\boldsymbol{A}$ with probability at least $1 - \delta$. Then, the number of matrix-vector queries must be at least $m = \Omega\left(\frac{\sqrt{\log(1/\delta)}}{\epsilon} + \frac{\log(1/\delta)}{\log(\log(1/\delta))}\right)$.*

## 1.2 Related Work

A summary of prior work on the query complexity of trace estimation of PSD matrices is given in **Table** 1. For the upper bounds, prior to the work of [30], the analysis of implicit trace estimation mainly focused on the variance of estimation with different types of query vectors. [30] gave the first upper bound on the query complexity. The work of [15] improved the bounds in [30]. On the lower bound side, although [15] gives a necessary condition on the query complexity for Gaussian query vectors, this condition does not directly translate to a bound on the minimum number of query vectors. The work of [16] gives the first lower bound on the query complexity in terms of $\epsilon$ but only works for constant failure probability.

| Upper Bounds | | | | |
|---|---|---|---|---|
| Prior Work | Query Complexity | Query Vector Type | Failure Probability | Algorithm Type |
| [30] | $O(\log(1/\delta)/\epsilon^2)$ | Gaussian | $\delta$ | non-adaptive |
| [30] | $O(\log(\mathrm{rank}(\boldsymbol{A})/\delta)/\epsilon^2)$ | Rademacher | $\delta$ | non-adaptive |
| [15] | $O(\log(1/\delta)/\epsilon^2)$ | Gaussian, Rademacher | $\delta$ | non-adaptive |
| [16] | $O(\sqrt{\log(1/\delta)}/\epsilon + \log(1/\delta))$ | Gaussian, Rademacher | $\delta$ | adaptive |
| [16] | $O(\log(1/\delta)/\epsilon)$ | Gaussian, Rademacher | $\delta$ | non-adaptive |
| **This Work** | $O(\sqrt{\log(1/\delta)}/\epsilon + \log(1/\delta))$ | Gaussian | $\delta$ | non-adaptive |
| Lower Bounds | | | | |
| [16] | $\Omega(1/(\epsilon \log(1/\epsilon)))$ | — | constant | adaptive |
| [16] | $\Omega(1/\epsilon)$ | — | constant | non-adaptive |
| **This Work** | $\Omega(\sqrt{\log(1/\delta)}/\epsilon + \frac{\log(1/\delta)}{\log \log(1/\delta)})$ | — | $\delta$ | non-adaptive |

Table 1: Upper and lower bounds on the query complexity for trace estimation of PSD matrices.

## 2 Problem Setting

**Notation.** A matrix $A \in \mathbb{R}^{n \times n}$ is symmetric positive semi-definite (PSD) if it is real, symmetric and has non-negative eigenvalues. Hence, $x^\top A x \geq 0$ for all $x \in \mathbb{R}^n$. Let $\text{tr}(A) = \sum_{i=1}^n A_{ii}$ denote the trace of $A$. Let $\|A\|_F = (\sum_{i=1}^n \sum_{j=1}^n A_{ij}^2)^{1/2}$ denote the Frobenius norm and $\|A\|_{op} = \sup_{\|\mathbf{v}\|_2 = 1} \|A\mathbf{v}\|_2$ denote the operator norm of $A$. Let $\mathcal{N}(\mu, \sigma^2)$ denote the Gaussian distribution with mean $\mu$ and variance $\sigma^2$. Our analysis extensively relies on the following facts:

**Definition 2.1** (Gaussian and Wigner Random Matrices). *We let $G \sim \mathcal{N}(n)$ denote an $n \times n$ random Gaussian matrix with i.i.d. $\mathcal{N}(0,1)$ entries. We let $W \sim \mathcal{W}(n) = G + G^T$ denote an $n \times n$ Wigner matrix, where $G \sim \mathcal{N}(n)$.*

**Fact 2.1** (Rotational Invariance of a standard Gaussian). *Let $R \in \mathbb{R}^{n \times n}$ be an orthornormal matrix. Let $\mathbf{g} \in \mathbb{R}^n$ be a random vector with i.i.d. $\mathcal{N}(0,1)$ entries. Then $R\mathbf{g}$ has the same distribution as $\mathbf{g}$.*

**Fact 2.2** (Upper and Lower Gaussian Tail Bounds). *Letting $Z \sim \mathcal{N}(0,1)$ be a univariate Gaussian random variable, for any $t > 0$, $\Pr[|Z| \geq t] = \Theta(t^{-1} \exp(-\frac{t^2}{2}))$.*

## 3 An Improved Analysis of `NA-Hutch++`

Suppose we are trying to compute a sketch so as to estimate the trace of a matrix $A$ up to a $(1 \pm \epsilon)$-factor with success probability at least $1 - \delta$. Note that we focus on the case where we make matrix-vector queries *non-adaptively*. For any algorithm that accomplishes this with small constant failure probability, one can simply repeat this procedure $O(\log(1/\delta))$ times to amplify the success probability to $1 - \delta$. Since these queries are non-adaptive and must be presented before any observations are made, it seems intuitive that the number of non-adaptive queries of `NA-Hutch++` (**Algorithm** 2) should be $O(\log(1/\delta)/\epsilon)$ as shown in [16]. In this section, we give a proof sketch as to why this can be reduced to $O(\sqrt{\log(1/\delta)}/\epsilon + \log(1/\delta))$ as stated in **Theorem** 3.1. All proof details are provided in the supplementary material.

**Theorem 3.1.** *Let $A$ be a PSD matrix. If `NA-Hutch++` is implemented with $m = O(\sqrt{\log(1/\delta)}/\epsilon + \log(1/\delta))$ matrix-vector multiplication queries, then with probability $1 - \delta$, the output of `NA-Hutch++`, denoted by $t$, satisfies $(1 - \epsilon)\text{tr}(A) \leq t \leq (1 + \epsilon)\text{tr}(A)$.*

`NA-Hutch++` splits its matrix-vector queries between computing an $O(1)$-approximate rank-$k$ approximation $\tilde{A}$ and performing Hutchinson's estimate on the residual matrix $A - \tilde{A}$ containing the small eigenvalues. The trade-off between the rank $k$ and the number $l$ of queries spent on estimating the small eigenvalues is summarized in **Theorem** 3.2.

**Theorem 3.2** (Theorem 4 of [16]). *Let $A \in \mathbb{R}^{n \times n}$ be PSD, $\delta \in (0, \frac{1}{2})$, $l \in \mathbb{N}, k \in \mathbb{N}$. Let $\tilde{A}$ and $\Delta$ be any matrices with $\text{tr}(A) = \text{tr}(\tilde{A}) + \text{tr}(\Delta)$ and $\|\Delta\|_F \leq O(1)\|A - A_k\|_F$ where $A_k = \arg\min_{\text{rank } k \ A_k} \|A - A_k\|_F$. Let $H_l(M)$ denote Hutchinson's trace estimator with $l$ queries on matrix $M$. For fixed constants $c, C$, if $l \geq c\log(\frac{1}{\delta})$, then with probability $1 - \delta$, for $Z = \text{tr}(\tilde{A}) + H_l(\Delta)$, we have $|Z - \text{tr}(A)| \leq C\sqrt{\frac{\log(1/\delta)}{kl}} \cdot \text{tr}(A)$.*

The total number of matrix-vector queries directly depends on the number of non-adaptive queries required to compute an $O(1)$-approximate rank-$k$ approximation $\tilde{A}$. Consider $S \in \mathbb{R}^{n \times c_1 m}, R \in \mathbb{R}^{n \times c_2 m}$ for some constants $c_1, c_2 > 0$ as defined in **Algorithm** 2, and set our low rank approximation of $A$ to be $\tilde{A} = AR(S^T AR)^\dagger (AS)^T$. The standard analysis [16] applies a result from streaming low-rank approximation in [27], which requires $m = O(k \log(1/\delta))$ to get $\|A - \tilde{A}\|_F \leq O(1)\|A - A_k\|_F$ with probability $1 - \delta$. [16] then sets $k = O(1/\epsilon)$ and $l = O(\log(1/\delta)/\epsilon)$ in **Theorem** 3.2 to get a $(1 \pm \epsilon)$ approximation to $\text{tr}(A)$. However, the right-hand side of **Theorem** 3.2 suggests the optimal split between $k$ and $l$ should be $k = l$. The reason [16] cannot achieve such an optimal split is due to a large number $m$ of queries to compute the $O(1)$-approximate rank $k$-approximation. We give an improved analysis of this result, which may be of independent interest.

To get $O(1)$ low rank approximation error, we need the non-adaptive query matrices $S, R$ to satisfy two properties: the subspace embedding property (see **Lemma** 3.3), and an approximate matrix product for orthogonal subspaces (see **Lemma** 3.4). While it is known that $m = O(k + \log(1/\delta))$

suffices to achieve the first property, we show that $m = O(k + \log(1/\delta))$ suffices to achieve the second property when $\boldsymbol{S}, \boldsymbol{R}$ are matrices with i.i.d. Gaussian random variables, stated in **Lemma** 3.4.

**Lemma 3.3** (Subspace Embedding (Theorem 6 of [28])). *Given $\delta \in (0, \frac{1}{2})$ and $\epsilon \in (0, 1)$. Let $\boldsymbol{S} \in \mathbb{R}^{r \times n}$ be a random matrix with i.i.d. Gaussian random variables $\mathcal{N}(0, \frac{1}{r})$. Then for any fixed $d$-dimensional subspace $\boldsymbol{A} \in \mathbb{R}^{n \times d}$, and for $r = O((d + \log(\frac{1}{\delta}))/\epsilon^2)$, the following holds with probability $1 - \delta$ simultaneously for all $x \in \mathbb{R}^d$, $\|\boldsymbol{SA}x\|_2 = (1 \pm \epsilon)\|\boldsymbol{A}x\|_2$*

**Lemma 3.4** (Approximate Matrix Product for Orthogonal Subspaces). *Given $\delta \in (0, \frac{1}{2})$, let $\boldsymbol{U} \in \mathbb{R}^{n \times k}, \boldsymbol{W} \in \mathbb{R}^{n \times p}$ be two matrices with orthonormal columns such that $\boldsymbol{U}^T \boldsymbol{W} = 0$, $p \geq \max(k, \log(1/\delta))$, $rank(\boldsymbol{U}) = k$ and $rank(\boldsymbol{W}) = p$. Let $\boldsymbol{S} \in \mathbb{R}^{r \times n}$ be a random matrix with i.i.d. Gaussian random variables $\mathcal{N}(0, \frac{1}{r})$. For $r = O(k + \log(\frac{1}{\delta}))$, the following holds with probability $1 - \delta$, $\|\boldsymbol{U}^T \boldsymbol{S}^T \boldsymbol{SW}\|_F \leq O(1)\|\boldsymbol{W}\|_F$.*

Note that we will apply the above two lemmas with constant $\epsilon$. The proof intuition is as follows: consider a sketch matrix $\boldsymbol{S}$ of size $r$ with i.i.d. $\mathcal{N}(0, \frac{1}{r})$ random variables as in **Lemma** 3.4. The range of $\boldsymbol{U} \in \mathbb{R}^{n \times k}$ corresponds to an orthonormal basis of a rank-$k$ low rank approximation to $\boldsymbol{A}$, and the range of $\boldsymbol{W} \in \mathbb{R}^{n \times p}$ is the orthogonal complement. Note that both $\boldsymbol{SU}$ and $\boldsymbol{SW}$ are random matrices consisting of i.i.d. $\mathcal{N}(0, \frac{1}{r})$ random variables and thus the task is to bound the size, in Frobenius norm, of the product of two random Gaussian matrices with high probability. Intuitively, the size of the matrix product is proportional to the rank $k$ and inversely proportional to our sketch size $r$. The overall failure probability $\delta$, however, is inversely proportional to $k$, since as $k$ grows, the matrix product involves summing over more squared Gaussian random variables, i.e., $\chi^2$ random variables, and thus becomes even more concentrated. We show that for $k \geq \log(1/\delta)$, a sketch size of $O(k)$ suffices since the failure probability for each $\chi^2$ random variable is small enough to pay a union bound over $k$ terms. On the other hand, when $k < \log(1/\delta)$, we show that $r = O(\log(1/\delta))$ suffices for the union bound. Combining the two cases gives $r = O(k + \log(1/\delta))$.

Having shown the above, we next show that the low rank approximation error, i.e., $\|\boldsymbol{A} - \tilde{\boldsymbol{A}}\|_F$, is upper bounded by: 1) the inflation in eigenvalues by applying a sketch matrix $\boldsymbol{S}$ as in **Lemma** 3.3; and 2) the approximate product of the range of a low rank approximation to $\boldsymbol{A}$ and its orthogonal complement, as in **Lemma** 3.4. Together these show that $m = O(k + \log(1/\delta))$ suffices for $\tilde{\boldsymbol{A}}$ to be an $O(1)$-approximate rank-$k$ approximation to $\boldsymbol{A}$ with probability $1 - \delta$, as stated in **Theorem** 3.5. Note that in both **Lemma** 3.3 and **Lemma** 3.4, the entries of the random matrix are scaled Gaussian random variables $\mathcal{N}(0, \frac{1}{r})$. However, when one sets the low rank approximation as $\tilde{\boldsymbol{A}} = \boldsymbol{AR}(\boldsymbol{S}^T \boldsymbol{AR})^\dagger (\boldsymbol{AS})^T$, the scale cancels and one can choose standard Gaussians in the sketching matrix for convenience as in **Theorem** 3.5.

**Theorem 3.5.** *Let $\boldsymbol{A} \in \mathbb{R}^{n \times n}$ be an arbitrary PSD matrix. Let $\boldsymbol{A}_k = \arg\min_{rank\text{-}k A_k} \|A - A_k\|_F$ be the optimal rank-$k$ approximation to $\boldsymbol{A}$ in Frobenius norm. If $\boldsymbol{S} \in \mathbb{R}^{n \times m}$ and $\boldsymbol{R} \in \mathbb{R}^{n \times cm}$ are random matrices with i.i.d. $\mathcal{N}(0, 1)$ entries for some fixed constant $c > 0$ with $m = O(k + \log(1/\delta))$, then with probability $1 - \delta$, the matrix $\widetilde{\boldsymbol{A}} = (\boldsymbol{AR})(\boldsymbol{S}^T \boldsymbol{AR})^\dagger (\boldsymbol{AS})^T$ satisfies $\|\boldsymbol{A} - \widetilde{\boldsymbol{A}}\|_F \leq O(1)\|\boldsymbol{A} - \boldsymbol{A}_k\|_F$.*

This improved result enables us to choose $k = l = O(\sqrt{\log(1/\delta)}/\epsilon)$ in **Theorem** 3.2, and combined with **Theorem** 3.5, this shows that only $O(\sqrt{\log(1/\delta)}/\epsilon + \log(1/\delta))$ matrix-vector queries are needed to output a number in $(1 \pm \epsilon)\text{tr}(\boldsymbol{A})$ with probability $1 - \delta$, as we conclude in **Theorem** 3.1.

## 4 Lower Bounds

In this section, we show that our upper bound on the query complexity of non-adaptive trace estimation is tight, up to a factor of $O(\log\log(1/\delta))$.

**Theorem 4.1** (Lower Bound for Non-Adaptive Queries). *Let $\epsilon \in (0, 1)$. Any algorithm that accesses a real PSD matrix $\boldsymbol{A}$ through matrix-vector multiplication queries $\boldsymbol{A}\mathbf{q}_1, \boldsymbol{A}\mathbf{q}_2, \dots, \boldsymbol{A}\mathbf{q}_m$, where $\mathbf{q}_1, \dots, \mathbf{q}_m$ are real-valued, non-adaptively chosen vectors, requires $m = \Omega\left(\frac{\sqrt{\log(1/\delta)}}{\epsilon} + \frac{\log(1/\delta)}{\log\log(1/\delta)}\right)$ queries to output an estimate $t$ such that with probability at least $1 - \delta$, $(1 - \epsilon)\text{tr}(\boldsymbol{A}) \leq t \leq (1 + \epsilon)\text{tr}(\boldsymbol{A})$.*

Our lower bound hinges on two separate cases: we first show an $\Omega(\sqrt{\log(1/\delta)}/\epsilon)$ lower bound in Section 4.1 whenever $\epsilon = O(1/\sqrt{\log(1/\delta)})$. Second, we show an $\Omega(\frac{\log(1/\delta)}{\log\log(1/\delta)})$ lower bound in Section 4.2 that applies to any $\epsilon \in (0,1)$. Observe that for $\epsilon < 1/\sqrt{\log(1/\delta)}$, the first lower bound holds; for $\epsilon \geq 1/\sqrt{\log(1/\delta)}$, our second lower bound dominates. Therefore, combining both lower bounds implies that for every $\epsilon$ and $\delta$, the query complexity of $O(\sqrt{\log(1/\delta)}/\epsilon + \log(1/\delta))$ for non-adaptive trace estimation is tight, up to a $\log\log(1/\delta)$ factor.

We now give a proof sketch of the two lower bounds. All details are in the supplementary material. Our lower bounds crucially make use of rotational invariance of the Gaussian distribution (see Fact 2.1) to argue that the first $q$ queries are, w.l.o.g., the standard basis vectors $e_1, ..., e_q$. Note that our queries can be assumed to be orthonormal. Both lower bounds use the family of $n \times n$ Wigner matrices (see Definition 2.1) with shifted mean, i.e., $\boldsymbol{W} + C \cdot \boldsymbol{I}$ for some $C > 0$ depending on $\|\boldsymbol{W}\|_{op}$, as part of the hard input distribution. The mean shift ensures that our ultimate instance is PSD with high probability.

## 4.1 Case 1: Lower Bound for Small $\epsilon$

The first lower bound is based on the observation that due to rotational invariance, the not-yet-queried part of $\boldsymbol{W}$ is distributed almost identically to $\boldsymbol{W}$, up to some mean shift, conditioned on the queried known part, no matter how the queries are chosen. The sum of diagonal entries of the not-yet-queried part is Gaussian, and this still has too much deviation to determine the overall trace of the input up to a $(1 \pm \epsilon)$ factor when $n = \sqrt{\log(1/\delta)}/\epsilon$ and $\epsilon < 1/\sqrt{\log(1/\delta)}$.

**Theorem 4.2** (Lower Bound for Small $\epsilon$). *For any PSD matrix $\boldsymbol{A}$ and all $\epsilon = O(1/\sqrt{\log(1/\delta)})$, any algorithm that succeeds with probability at least $1 - \delta$ in outputting an estimate $t$ such that $(1 - \epsilon)\mathrm{tr}(\boldsymbol{A}) \leq t \leq (1 + \epsilon)\mathrm{tr}(\boldsymbol{A})$, requires $m = \Omega(\sqrt{\log(1/\delta)}/\epsilon)$ matrix-vector queries.*

## 4.2 Case 2: Lower Bound for Every $\epsilon$

The second lower bound presented in **Theorem** 4.3 is shown via reduction to a distribution testing problem between two distributions presented in **Problem** 4.4.

**Theorem 4.3** (Lower Bound on Non-adaptive Queries for PSD Matrices). *Let $\epsilon \in (0,1)$. Any algorithm that accesses a real, PSD matrix $\boldsymbol{A}$ through matrix-vector queries $\boldsymbol{A}\mathbf{q}_1, \boldsymbol{A}\mathbf{q}_2, \ldots, \boldsymbol{A}\mathbf{q}_m$, where $\mathbf{q}_1, \ldots, \mathbf{q}_m$ are real-valued non-adaptively chosen vectors, requires $m = \Omega(\frac{\log(1/\delta)}{\log\log(1/\delta)})$ to output an estimate $t$ such that with probability at least $1 - \delta$, $(1 - \epsilon)\mathrm{tr}(\boldsymbol{A}) \leq t \leq (1 + \epsilon)\mathrm{tr}(\boldsymbol{A})$.*

In the distribution testing problem, we consider Wigner matrices $\boldsymbol{W} \sim \mathcal{W}(\log(1/\delta))$ shifted by $\Theta(\sqrt{\log(1/\delta)})\boldsymbol{I}$. The problem requires an algorithm for distinguishing between a sample $\mathcal{Q}$ from this Wigner distribution and a sample $\mathcal{P}$ from this distribution shifted by a random rank-1 PSD matrix. The rank-1 matrix is the outer product of a random vector with itself and is chosen to provide a constant factor gap between the trace of $\mathcal{P}$ and $\mathcal{Q}$.

**Problem 4.4** (Hard PSD Matrix Distribution Test). *Given $\delta \in (0, \frac{1}{2})$, set $n = \log(1/\delta)$. Choose $\mathbf{g} \in \mathbb{R}^n$ to be an independent random vector with i.i.d. $\mathcal{N}(0,1)$ entries. Consider two distributions:*

- *Distribution $\mathcal{P}$ on matrices $\left\{ C \log^{3/2}(\frac{1}{\delta}) \cdot \frac{1}{\|\mathbf{g}\|_2^2} \mathbf{g}\mathbf{g}^T + \boldsymbol{W} + 2\sqrt{\log(\frac{1}{\delta})}\boldsymbol{I} \right\}$, for some fixed constant $C > 1$.*

- *Distribution $\mathcal{Q}$ on matrices $\left\{ \boldsymbol{W} + 2\sqrt{\log(\frac{1}{\delta})}\boldsymbol{I} \right\}$.*

*where $\boldsymbol{W} \sim \mathcal{W}(n)$ as in Definition 2.1. Let $\boldsymbol{A}$ be a random matrix drawn from either $\mathcal{P}$ or $\mathcal{Q}$ with equal probability. Consider any algorithm which, for a fixed query matrix $\boldsymbol{Q} \in \mathbb{R}^{n \times q}$, observes $\boldsymbol{A}\boldsymbol{Q}$, and guesses if $\boldsymbol{A} \sim \mathcal{P}$ or $\boldsymbol{A} \sim \mathcal{Q}$ with success probability at least $1 - \delta$.*

We then show in **Lemma** 4.5 that any algorithm which succeeds with probability $1 - \delta$ in distinguishing $\mathcal{P}$ from $\mathcal{Q}$ requires $\Omega(\frac{\log(1/\delta)}{\log\log(1/\delta)})$ non-adaptive matrix-vector queries.

Due to rotational invariance and since queries are non-adaptive, the first $q$ queries are the first $q$ standard unit vectors. By Fact 2.2, with probability at least $\frac{1}{\log(1/\delta)}$, however, a single coordinate of $\mathbf{g}$

has absolute value at most $\frac{1}{\log(1/\delta)}$. By independence, with probability at least $(\frac{1}{\log(1/\delta)})^q$, all of the first $q$ coordinates of $\mathbf{g}$ are simultaneously small, and thus give the algorithm almost no information to distinguish $\mathcal{P}$ from $\mathcal{Q}$; this probability is $\delta$ if $q = O(\frac{\log(1/\delta)}{\log\log(1/\delta)})$.

**Lemma 4.5** (Hardness of Problem 4.4). *For a non-adaptive query matrix $\boldsymbol{Q} \in \mathbb{R}^{n \times q}$ as in **Problem 4.4**, given $\delta \in (0, \frac{1}{2})$, for $n = \log(1/\delta)$, if $q = o(\frac{\log(1/\delta)}{\log\log(1/\delta)})$, no algorithm can solve **Problem 4.4** with success probability $1 - \delta$.*

## 5 Experiments

[1]**Part I: Comparison of Failure Probability and Running Time** We give sequential and parallel implementations of the non-adaptive trace estimation algorithm `NA-Hutch++` (**Algorithm** 2), the adaptive algorithm `Hutch++` (**Algorithm** 1) and Hutchinson's method [14]. We specifically explore the benefits of the non-adaptive algorithm in a parallel setting, where all algorithms have parallel access to a matrix-vector oracle. All the code is included in the supplementary material and will be publicly released.

**Metrics.** We say an estimate failed if on input matrix $\boldsymbol{A}$, the estimate $t$ returned by an algorithm falls into either case: $t < (1 - \epsilon)\mathrm{tr}(\boldsymbol{A})$ or $t > (1 + \epsilon)\mathrm{tr}(\boldsymbol{A})$. We measure the performance of each algorithm by: 1) the number of failed estimates across 100 random trials, 2) the total wall-clock time to perform 100 trials with sequential execution, and 3) the total wall-clock time to perform 100 trials with parallel execution.

**Datasets and Applications.** We consider different applications of trace estimation from synthetic to real-world datasets. In many applications, trace estimation is used to estimate not only $\mathrm{tr}(\boldsymbol{A})$, but also $\mathrm{tr}(f(\boldsymbol{A}))$ for some function $f : \mathbb{R} \to \mathbb{R}$. Letting $\boldsymbol{A} = \boldsymbol{V}\boldsymbol{\Sigma}\boldsymbol{V}^T$ be the eigendecomposition of $\boldsymbol{A}$, we have $f(\boldsymbol{A}) := \boldsymbol{V}f(\boldsymbol{\Sigma})\boldsymbol{V}^T$, where $f(\boldsymbol{\Sigma})$ denotes applying $f$ to each of the eigenvalues. Due to the expensive computation of eigendecompositions of large matrices, the matrix-vector multiplication $f(\boldsymbol{A})\mathbf{v}$ is often estimated by polynomials implicitly computed via an oracle algorithm for a random vector $\mathbf{v}$. The Lanczos algorithm is a very popular choice due to its superior performance (e.g. [31, 2, 7]). We compare the performance of our trace estimation algorithms on the following applications and datasets, and use the Lanczos algorithm as the matrix-vector oracle on a random vector $\mathbf{v}$ in some particular cases.

- **Fast Decay Spectrum.** We first consider a `synthetic` dataset of size 5000 with a fast decaying spectrum, following [16], which is a diagonal matrix $\boldsymbol{A}$ with $i$-th diagonal entry $\boldsymbol{A}_{ii} = 1/i^2$. Matrices with fast decaying spectrum will cause high variance in the estimated trace of `Huthinson`, but low variance for `Hutch++` and `NA-Hutch++`. The matrix-vector oracle is simply $\boldsymbol{A}\mathbf{v}$.

- **Graph Estrada Index.** Given a binary adjacency matrix $\boldsymbol{A} \in \{0, 1\}^{n \times n}$ of a graph, the Graph Estrada Index is defined as $\mathrm{tr}(\exp(\boldsymbol{A}))$, which measures the strength of connectivity within the graph. Following [16], we use `roget`'s Thesaurus semantic graph[2] with 1022 nodes, which was originally studied in [5], and use the Lanczos algorithm with 40 steps to approximate $\exp(\boldsymbol{A})\mathbf{v}$ as the matrix-vector oracle.

- **Graph Triangle Counting.** Given a binary adjacency matrix $\boldsymbol{A} \in \{0, 1\}^{n \times n}$ of a graph, the number of triangles in the graph is $1/6 \cdot \mathrm{tr}(\boldsymbol{A}^3)$. This is an important graph summary with numerous applications in graph-mining and social network analysis (e.g. [32, 33]). We use `arxiv_cm`, the Condense Matter collaboration network dataset from arXiv [3]. This is a common benchmark graph with $23, 133$ nodes and $173, 361$ triangles. The matrix-vector oracle is $\boldsymbol{A}^3\mathbf{v}$. Note that $\boldsymbol{A}^3$ in this case is not necessarily a PSD matrix.

- **Log-likelihood Estimation for Gaussian Process.** When performing maximum likelihood estimation (MLE) to optimize the hyperparameters of a kernel matrix $\boldsymbol{A}$ for Gaussian Processes, one needs to compute the gradient of the log-determininant of $\boldsymbol{A}$, which involves estimating $\mathrm{tr}(\boldsymbol{A}^{-1})$ [2]. Following [2], we use the `precipitation`[4] dataset, which consists

---

[1]Our code is available at: https://github.com/11hifish/OptSketchTraceEst

[2]http://vlado.fmf.uni-lj.si/pub/networks/data/

[3]https://snap.stanford.edu/data/ca-CondMat.html

[4]https://catalog.data.gov/dataset/u-s-hourly-precipitation-data

of the measured amount of precipitation during a day collected from 5,500 weather stations in the US in 2010. We sample 1,000 data points, and construct a covariance matrix $A$ using the RBF kernel with length scale 1. We use the Lanczos algorithm with 40 steps as in [2] to approximate $A^{-1}v$ as the matrix-vector oracle.

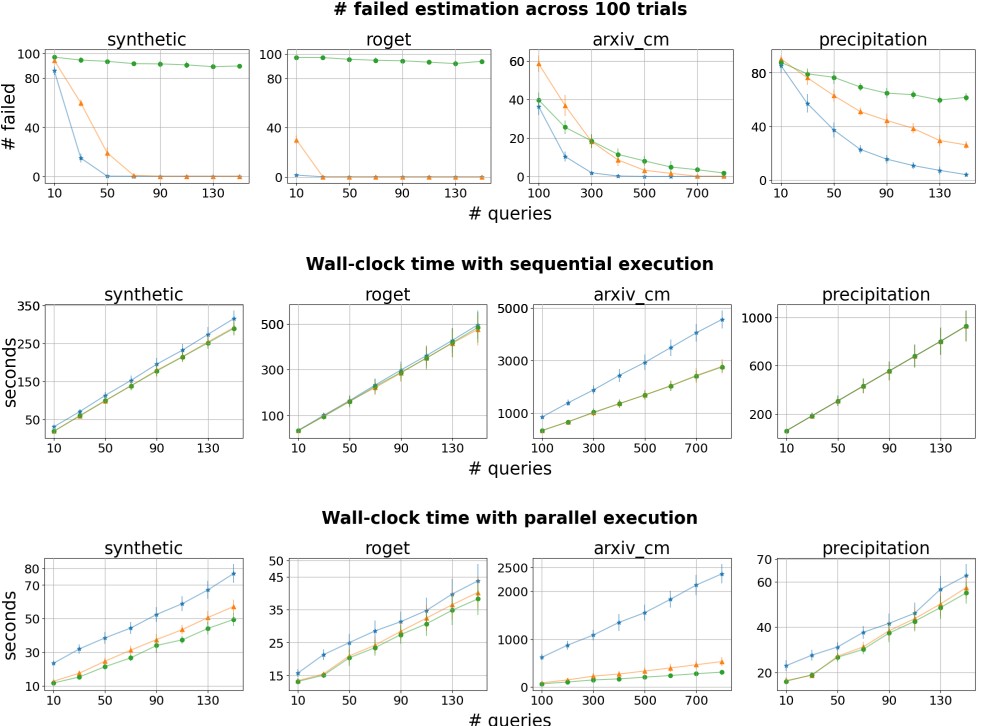

Figure 1: The performance comparison of Hutch++, NA-Hutch++ and Huthinson over 4 datasets (mean $\pm$ 1 std. across 10 random runs). The approximation error for all settings is set at $\epsilon = 0.01$. Both Hutch++ and NA-Hutch++ outperform Hutchinson in terms of failed estimates. The parallel version of the non-adaptive NA-Hutch++ is significantly faster than the adaptive Hutch++, making it more practical in real-world applications. *Legend:* Hutch++ is —★—, NA-Hutch++ is —▲—, and Hutchinson is —●—.

**Implementation.** We use random vectors with i.i.d. $\mathcal{N}(0, 1)$ entries as the query vectors for all algorithms. NA-Hutch++ requires additional hyperparameters to specify how the queries are split between random matrices $S, R, G$ (see **Algorithm** 2). We set $c_1 = c_3 = \frac{1}{4}$ and $c_2 = \frac{1}{2}$ as [16] suggests. For each setting, we conduct 10 random runs and report the mean number of failed estimates across 100 trials and the mean total wall-clock time (in seconds) conducting 100 trials with one standard deviation. For all of our experiments, we fix the error parameter $\epsilon = 0.01$ and measure the performance of each algorithm with $\{10, 30, 50, \ldots, 130, 150\}$ queries on synthetic, roget and precipitation, and with $\{100, 200, \ldots, 700, 800\}$ queries on arxiv_cm which has a significantly larger size. The parallel versions are implemented using Python multiprocessing[5] package. Due to the large size of arxiv_cm, we use sparse_dot_mkl[6], a Python wrapper for Intel Math Kernel Library (MKL) which supports fast sparse matrix-vector multiplications, to implement the matrix-vector oracle for this dataset. During the experiments, we launch a pool of 40 worker processes in our parallel execution. All experiments are conducted on machines with 40 CPU cores.

**Results and Discussion.** The results of Hutch++, NA-Hutch++ and Hutchinson over the 4 datasets are presented in **Figure** 1. The performance of all algorithms is consistent across different datasets with different matrix-vector oracles, and even on a non-PSD instance from arxiv_cm. Given the same number of queries, Hutch++ and NA-Hutch++ both give significantly fewer failed estimates than Hutchinson, particularly on PSD instances. It is not surprising to see that Hutchinson fails to

---

[5] https://docs.python.org/3/library/multiprocessing.html
[6] https://github.com/flatironinstitute/sparse_dot

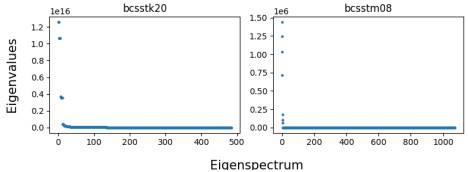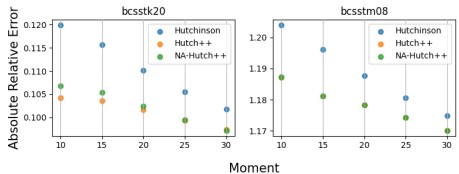

Figure 2: The eigenspectrum of the two datasets and the performance comparison of Hutch++, NA-Hutch++ and Hutchinson on maximum entropy estimation based log determinant estimation.

achieve a $(1 \pm \epsilon)$-approximation to the trace most of the time due to the high variance in its estimation, given a small number of queries and a high accuracy requirement ($\epsilon = 0.01$).

For computational costs, the difference in running time of all algorithms is insignificant in our sequential execution. In our parallel execution, however, `Hutch++` becomes significantly slower than the other two, `NA-Hutch++` and `Hutchinson`, which have very little difference in their parallel running time. `Hutch++` suffers from slow running time due to its adaptively chosen queries, despite the fact that `Hutch++` consistently gives the least number of failed estimates.

It is not hard to see that `NA-Hutch++` gives the best trade-off between a high success probability in estimating an accurate trace with only a few number of queries, and a fast parallel running time due to the use of non-adaptive queries, which makes `NA-Hutch++` more practical on large, real-world datasets. We remark that although the Lanczos algorithm is adaptive itself, even with a sequential matrix-vector oracle, our non-adaptive trace estimation can still exploit much more parallelism than adaptive methods, as shown by our experiments.

**Part II: Comparison of Performance on Log Determinant Estimation** We give an additional experiment to compare the performance of `Hutch++`, `NA-Hutch++` and `Hutchinson` on estimating $\log(\det(\mathbf{K})) = \mathrm{tr}(\log(\mathbf{K}))$, for some covariance matrix $\mathbf{K}$. Estimating $\log(\det(\mathbf{K}))$ is required when computing the marginal log-likelihood in large-scale Gaussian Process models. Recently, [3] proposed a maximum entropy estimation based method for log determinant estimation, which uses Hutchinson's trace estimation as a subroutine to estimate up to the $k$-th moments of the eigenvalues, given a fixed $k$. The $i$-th moment of the eigenvalues is $\mathbb{E}[\lambda^i] = \frac{1}{n}\mathrm{tr}(\mathbf{K}^i)$, where $\mathbf{K}$ is an $n \times n$ PSD matrix, and $\lambda$ is the vector of eigenvalues. [3] shows that their proposed approach outperforms traditional Chebyshev/Lanczos polynomials for computing $\log(\det(\mathbf{K}))$ in terms of absolute value of the relative error, i.e., abs (estimated log determinant - true log determinant)/abs(true log determinant).

We compare the estimated log determinant of a covariance matrix with different trace estimation subroutines for estimating the moments of the eigenvalues. We use 2 PSD matrices from the UFL Sparse Matrix Collection[7]: `bcsstk20` (size $485 \times 485$) and `bcsstm08` (size $1074 \times 1074$), with varying max moments $\{10, 15, \dots, 30\}$ and 30 matrix-vector queries. We repeated each run 100 times and reported the mean estimated log determinant with each trace estimation subroutine. While an improved estimate of the eigenvalue moments does not necessarily lead to an improved estimate of the log determinant, it is not hard to show that an accurate moment estimation does lead to improved log determinant estimation in extreme cases where the eigenspectrum of $\mathbf{K}$ contains a few very large eigenvalues. Such a case will cause Hutchinson's method to have very large variance, while our method reduces the variance by first removing the large eigenvalues. The eigenspectrums of both input matrices and the results are presented in **Figure** 2.

## 6 Conclusion

We determine an optimal $\Theta(\sqrt{\log(1/\delta)}/\epsilon + \log(1/\delta))$ bound on the number of queries to achieve $(1 \pm \epsilon)$ approximation of the trace with probability $1 - \delta$ for non-adaptive trace estimation algorithms, up to a $\log\log(1/\delta)$ factor. This involves both designing a new algorithm, as well as proving a new lower bound. We conduct experiments on synthetic and real-world datasets and confirm that our non-adaptive algorithm has a higher success probability compared to Hutchinson's method for the same sketch size, and has a significantly faster parallel running time compared to adaptive algorithms.

---

[7] https://sparse.tamu.edu/

## Acknowledgments and Disclosure of Funding

We would like to thank the anonymous reviewers for their feedback. We are also grateful to Raphael Meyer for many detailed comments on the lower bound proofs. D. Woodruff was supported by NSF CCF-1815840, Office of Naval Research grant N00014-18-1-2562, and a Simons Investigator Award.

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
