# Appendices

## Contents

# A  Basic Facts about Gaussian Distributions

Let $\mathcal{N}(\mu, \sigma^2)$ denote a Gaussian distribution with mean $\mu$ and variance $\sigma^2$. Let $\chi^2(n)$ denote a $\chi^2$ distribution with $n$ degrees of freedom. Our analysis extensively uses the following facts about Gaussian and $\chi^2$ distributions:

**Definition A.1** (Gaussian and Wigner Random Matrices). *We let $\boldsymbol{G} \sim \mathcal{N}(n)$ denote an $n \times n$ random Gaussian matrix with i.i.d. $\mathcal{N}(0, 1)$ entries. We let $\boldsymbol{W} \sim \mathcal{W}(n) = \boldsymbol{G} + \boldsymbol{G}^T$ denote an $n \times n$ Wigner matrix, where $\boldsymbol{G} \sim \mathcal{N}(n)$.*

**Fact A.1** ($\chi^2$ Tail Bound (**Lemma 1** of [1])). *Let $Z \sim \chi^2(n)$. Then for any $x > 0$,*

$$\Pr[Z \geq n + 2\sqrt{nx} + 2x] \leq e^{-x}$$
$$\Pr[Z \leq n - 2\sqrt{nx}] \leq e^{-x}$$

**Fact A.2** (Rotational Invariance). *Let $\boldsymbol{R} \in \mathbb{R}^{n \times n}$ be an orthornormal matrix. Let $\mathbf{g} \in \mathbf{R}^n$ be a random vector with i.i.d. $\mathcal{N}(0, 1)$ entries. Then $\boldsymbol{R}\mathbf{g}$ has the same distribution as $\mathbf{g}$.*

**Fact A.3** (Upper Gaussian Tail Bound). *Let $Z \sim \mathcal{N}(0, \sigma^2)$ be a univariate Gaussian random variable. Then for any $t > 0$,*

$$\Pr[Z \geq t] \leq \exp(-\frac{t^2}{2\sigma^2})$$

**Fact A.4** (Lower Gaussian Tail Bound). *Letting $Z \sim \mathcal{N}(0, 1)$ be a univariate Gaussian random variable, for any $t > 0$,*

$$\Pr[Z \geq t] \geq \frac{1}{\sqrt{2\pi}} \cdot \frac{1}{t} \exp(t^2/2)$$

**Lemma A.2** (Concentration of Singular Values of a Gaussian Random Matrix (**Eq. 2.3** of [2])). *Let $\boldsymbol{G} \sim \mathcal{N}(n)$, and $s_{max}(\boldsymbol{G})$ denote the maximum singular value of $\boldsymbol{G}$. Then $\forall t \geq 0$,*

$$\Pr[s_{max}(\boldsymbol{G}) \leq 2\sqrt{n} + t] \geq 1 - 2\exp(-t^2/2)$$

**Fact A.5** (KL Divergence Between Multivariate Gaussian Distributions (**Eq. 8** of [3], or Section 9 of [4]). *Let $\mathcal{P} \sim \mathcal{N}(\mu_1, \boldsymbol{\Sigma}_1)$ and $\mathcal{Q} \sim \mathcal{N}(\mu_2, \boldsymbol{\Sigma}_2)$ be two $k$-dimensional multivariate normal distributions. The Kullback-Leibler divergence between $\mathcal{P}$ and $\mathcal{Q}$ is*

$$\mathcal{D}_{KL}(\mathcal{P} \parallel \mathcal{Q}) = \frac{1}{2}\Big\{(\mu_2 - \mu_1)^T \boldsymbol{\Sigma}_2^{-1}(\mu_2 - \mu_1) + \text{tr}(\boldsymbol{\Sigma}_2^{-1}\boldsymbol{\Sigma}_1) - \ln\frac{\det(\boldsymbol{\Sigma}_1)}{\det(\boldsymbol{\Sigma}_2)} - k\Big\}$$

**Fact A.6** (Conditioning Increases KL Divergence (**Theorem 2.2 - 5** of [5])). *Let $\mathcal{P}_{Y|X}$, $\mathcal{Q}_{Y|X}$ be two conditional probability distributions over spaces $X \in \mathcal{X}$ and $Y \in \mathcal{Y}$, let $\mathcal{P}_Y = \mathcal{P}_{Y|X}\mathcal{P}_X$ and $\mathcal{Q}_Y = \mathcal{Q}_{Y|X}\mathcal{P}_X$. Then,*

$$\mathcal{D}_{KL}(\mathcal{P}_Y \parallel \mathcal{Q}_Y) \leq \mathcal{D}_{KL}(\mathcal{P}_{Y|X} \parallel \mathcal{Q}_{Y|X} \mid \mathcal{P}_X) := \int \mathcal{D}_{KL}(\mathcal{P}_{Y|X=x} \parallel \mathcal{Q}_{Y|X=x})d\mathcal{P}_X$$

**Fact A.7** (KL Divergence Data Processing Inequality (Page 18 of [6])). *For any function $f$ and random variables $X$ and $Y$ on the same probability space, it holds that*

$$\mathcal{D}_{KL}(f(X) \parallel f(Y)) \leq \mathcal{D}_{KL}(X \parallel Y)$$

# B  An Improved Analysis of `NA-Hutch++`

In this section, we give an improved analysis of `NA-Hutch++`, showing that the query complexity of `NA-Hutch++` can be improved from $O(\log(1/\delta)/\epsilon)$, as shown in [7], to $O\left(\frac{\sqrt{\log(1/\delta)}}{\epsilon} + \log(1/\delta)\right)$ on PSD (positive semidefinite) input matrices $\boldsymbol{A}$, to get a $(1 \pm \epsilon)$ approximation to $\text{tr}(\boldsymbol{A})$ with probability $1 - \delta$. The `NA-Hutch++` algorithm is duplicated here for convenience as follows:

---
**Algorithm 1** `NA-Hutch++` [7]: Stochastic trace estimation with **non-adaptive** matrix-vector queries
---
1: **Input:**  Matrix-vector multiplication oracle for PSD matrix $\boldsymbol{A} \in \mathbb{R}^{n \times n}$. Number $m$ of queries.
2: **Output:**  Approximation to $\text{tr}(\boldsymbol{A})$.
3: Fix constants $c_1, c_2, c_3$ such that $c_1 < c_2$ and $c_1 + c_2 + c_3 = 1$.
4: Sample $\boldsymbol{S} \in \mathbb{R}^{n \times c_1 m}$, $\boldsymbol{R} \in \mathbb{R}^{n \times c_2 m}$, and $\boldsymbol{G} \in \mathbb{R}^{n \times c_3 m}$, with i.i.d. $\mathcal{N}(0,1)$ entries.
5: $\boldsymbol{Z} = \boldsymbol{AR}$, $\boldsymbol{W} = \boldsymbol{AS}$
6: **return**  $t = \text{tr}((\boldsymbol{S}^T \boldsymbol{Z})^\dagger (\boldsymbol{W}^T \boldsymbol{Z})) + \frac{1}{c_3 m}\left(\text{tr}(\boldsymbol{G}^T \boldsymbol{AG}) - \text{tr}(\boldsymbol{G}^T \boldsymbol{Z}(\boldsymbol{S}^T \boldsymbol{Z})^\dagger \boldsymbol{W}^T \boldsymbol{G})\right)$.

---

**Roadmap.**  Recall that `NA-Hutch++` splits its matrix-vector queries between computing an $O(1)$-approximate rank-$k$ approximation $\widetilde{\boldsymbol{A}}$ and performing Hutchinson's estimate on the residual matrix $\boldsymbol{A} - \widetilde{\boldsymbol{A}}$. The key to an improved query complexity of `NA-Hutch++` is on the analysis of the size of random Gaussian sketching matrices $\boldsymbol{S}, \boldsymbol{R}$ in Algorithm 1 that one needs to get an $O(1)$-approximate rank-$k$ approximation $\widetilde{\boldsymbol{A}}$ in the Frobenius norm. To get the desired rank-$k$ approximation, we need $\boldsymbol{S}$ and $\boldsymbol{R}$ to satisfy two properties: 1) subspace embedding as in **Lemma** 3.3 and 2) approximate matrix product for orthogonal subspaces as in **Lemma** 3.4. Specifically, we show in **Lemma** 3.4 that choosing $\boldsymbol{S}$ and $\boldsymbol{R}$ to be of size $O(k + \log(1/\delta))$ suffices to get the second property with probability $1 - \delta$.

After that, we show in **Lemma** B.1 that if a sketching matrix $\boldsymbol{S}$ satisfies the two properties mentioned above, with size $O(k + \log(1/\delta))$, one gets an $O(1)$-approximate low rank approximation with probability $1 - \delta$ when solving a sketched version of the regression problem $\min_{\boldsymbol{X}} \|\boldsymbol{S}^T (\boldsymbol{AX} - \boldsymbol{B})\|_F$ for fixed matrices $\boldsymbol{A}, \boldsymbol{B}$ with $\text{rank}(\boldsymbol{A}) = k$. **Lemma** B.1 serves as an intermediate step to construct an $O(1)$-approximate rank-$k$ approximation $\widetilde{\boldsymbol{A}}$ with $\boldsymbol{S}, \boldsymbol{R}$ having a size of only $O(k + \log(1/\delta))$ in **Theorem** 3.5.

Finally, we combine **Theorem** 3.2 from [7], which shows the trade-off between the rank $k$ and the number $l$ spent on estimating the small eigenvalues, and **Theorem** 3.5, which shows the number of non-adaptive queries one needs to get a desired rank-$k$ factor, to conclude in **Theorem** 3.1 that `NA-Hutch++` needs only $O\left(\frac{\sqrt{\log(1/\delta)}}{\epsilon} + \log(1/\delta)\right)$ non-adaptive queries, by setting $k = \frac{\sqrt{\log(1/\delta)}}{\epsilon}$.

**Lemma 3.3** (Subspace Embedding (Theorem 6 of [8])). *Given $\delta \in (0, \frac{1}{2})$ and $\epsilon \in (0,1)$, let $\boldsymbol{S} \in \mathbb{R}^{r \times n}$ be a random matrix with i.i.d. Gaussian random variables $\mathcal{N}(0, \frac{1}{r})$. Then for any fixed $d$-dimensional subspace $\boldsymbol{A} \in \mathbb{R}^{n \times d}$, and for $r = O((d + \log(\frac{1}{\delta}))/\epsilon^2)$, the following holds with probability $1 - \delta$ simultaneously for all $x \in \mathbb{R}^d$,*

$$\|\boldsymbol{S}\boldsymbol{A}x\|_2 = (1 \pm \epsilon)\|\boldsymbol{A}x\|_2$$

**Lemma 3.4** (Approximate Matrix Product for Orthogonal Subspaces). *Given $\delta \in (0, \frac{1}{2})$, let $\boldsymbol{U} \in \mathbb{R}^{n \times k}, \boldsymbol{W} \in \mathbb{R}^{n \times p}$ be two matrices with orthonormal columns such that $\boldsymbol{U}^T \boldsymbol{W} = 0$, $p \geq \max(k, \log(1/\delta))$, $\text{rank}(\boldsymbol{U}) = k$ and $\text{rank}(\boldsymbol{W}) = p$. Let $\boldsymbol{S} \in \mathbb{R}^{r \times n}$ be a random matrix with i.i.d. Gaussian random variables $\mathcal{N}(0, \frac{1}{r})$. For $r = O(k + \log(\frac{1}{\delta}))$, the following holds with probability $1 - \delta$,*

$$\|\boldsymbol{U}^T \boldsymbol{S}^T \boldsymbol{S}\boldsymbol{W}\|_F \leq O(1)\|\boldsymbol{W}\|_F$$

*Proof.* Let $\boldsymbol{G} = \sqrt{r}\boldsymbol{U}^T \boldsymbol{S}^T \in \mathbb{R}^{k \times r}$ and $\boldsymbol{H} = \sqrt{r}\boldsymbol{S}\boldsymbol{W} \in \mathbb{R}^{r \times p}$. Since both $\boldsymbol{U}$ and $\boldsymbol{W}$ have orthonormal columns, both $\boldsymbol{G}$ and $\boldsymbol{H}$ are random matrices with i.i.d. Gaussian random variables $\mathcal{N}(0,1)$. Furthermore, let $\mathbf{g}_i, \forall i \in [k]$ denote the $i$-th row of $\boldsymbol{G}$ and $\mathbf{h}_j, \forall j \in [p]$ denote the $j$-th column of $\boldsymbol{H}$.

$$\|\boldsymbol{U}^T \boldsymbol{S}^T \boldsymbol{S}\boldsymbol{W}\|_F^2 = \left\|\frac{1}{\sqrt{r}}\boldsymbol{G}\frac{1}{\sqrt{r}}\boldsymbol{H}\right\|_F^2$$
$$= \frac{1}{r^2}\sum_{i=1}^{k}\sum_{j=1}^{p}\langle \mathbf{g}_i, \mathbf{h}_j\rangle^2$$

$$= \frac{1}{r^2} \sum_{i=1}^{k} \sum_{j=1}^{p} \|g_i\|_2^2 \left\langle \frac{\mathbf{g}_i}{\|\mathbf{g}_i\|_2}, \mathbf{h}_j \right\rangle^2$$

$$= \frac{1}{r^2} \sum_{i=1}^{k} \|g_i\|_2^2 \left( \sum_{j=1}^{p} \langle \frac{\mathbf{g}_i}{\|\mathbf{g}_i\|_2}, \mathbf{h}_j \rangle^2 \right)$$

Since $\|\frac{\mathbf{g}_i}{\|\mathbf{g}\|_2}\|_2 = 1$, $\langle \frac{\mathbf{g}_i}{\|\mathbf{g}_i\|_2}, \mathbf{h}_j \rangle \sim \mathcal{N}(0,1)$. Thus,

$$\|\boldsymbol{U}^T \boldsymbol{S}^T \boldsymbol{S} \boldsymbol{W}\|_F^2 = \frac{1}{r^2} \sum_{i=1}^{k} \mathbf{c}_i \cdot \mathbf{d}_i$$

where $\mathbf{c}_i \sim \chi^2(r)$, $\mathbf{d}_i \sim \chi^2(p)$, $\forall i \in [k]$. Note that since $\boldsymbol{W}$ has orthonormal columns, $\|\boldsymbol{W}\|_F^2 = p$.

The number $r$ of rows our random sketch matrix $\boldsymbol{S}$ needs in order to obtain an upper bound on the product of random Gaussian matrices $\boldsymbol{SU}$ and $\boldsymbol{SW}$, up to a constant factor of $\|\boldsymbol{W}\|_F$, depends on the concentration of $\boldsymbol{SU}$ and $\boldsymbol{SW}$. Specifically, to apply the $\chi^2$ tail bound on some random variable $\mathbf{v} \sim \chi^2(d)$ from Fact A.1 and to get that $\mathbf{v}$ concentrates around $O(1)d$ with probability $1 - \delta$, the degree $d$ needs to be at least $\log(1/\delta)$. Since we require $p = \text{rank}(\boldsymbol{W}) \geq \log(1/\delta)$, $\boldsymbol{SW}$ is concentrated with high probability. The concentration of $\boldsymbol{SU}$ depends on $\text{rank}(\boldsymbol{U}) = k$. To upper bound $\|(\boldsymbol{SU})^T(\boldsymbol{SW})\|_F$, we consider two cases for $k$:

**Case I:** Consider the case when $k \geq \log(\frac{1}{\delta})$:

Since $p \geq k \geq \log(\frac{1}{\delta})$, by **Fact** A.1, $\forall i \in [k]$,

$$\Pr[\mathbf{d}_i \leq O(1)p] \geq 1 - e^{-O(k)}$$

Since $r = O(k + \log(1/\delta))$, by **Fact** A.1, $\forall i \in [k]$,

$$\Pr[\mathbf{c}_i \leq O(1)k] \geq 1 - e^{-O(k)}$$

By a union bound over $2k$ $\chi^2$ random variables,

$$\Pr\left[ \sum_{i=1}^{k} \mathbf{c}_i \cdot \mathbf{d}_i \leq O(1)k^2 p \right] \geq 1 - 2k \cdot e^{-O(k)}$$

Thus with probability $1 - O(\delta)$,

$$\|\boldsymbol{U}^T \boldsymbol{S}^T \boldsymbol{S} \boldsymbol{W}\|_F^2 = \frac{1}{r^2} \sum_{i=1}^{k} \mathbf{c}_i \cdot \mathbf{d}_i$$

$$\leq \frac{1}{r^2} O(1)k^2 p$$

$$= \frac{1}{r^2} O(1)k^2 \|\boldsymbol{W}\|_F^2$$

And so $r = O(k + \log(1/\delta))$ gives $\|\boldsymbol{U}\boldsymbol{S}^T \boldsymbol{S} \boldsymbol{W}\|_F \leq O(1)\|\boldsymbol{W}\|_F$ with probability $1 - \delta$.

**Case II:** Consider the case when $k < \log(\frac{1}{\delta})$.

Since $p \geq \log(\frac{1}{\delta})$, by **Fact** A.1, $\forall i \in [k]$,

$$\Pr[\mathbf{d}_i \leq O(1)p] \geq 1 - e^{-O(\log(1/\delta))}$$

Since $r = O(k + \log(1/\delta))$, by **Fact** A.1, $\forall i \in [k]$,

$$\Pr[\mathbf{c}_i \leq O(1)\log(1/\delta)] \geq 1 - e^{-O(\log(1/\delta))}$$

By a union bound over $2k$ $\chi^2$ random variables, for $k < \log(1/\delta)$

$$\Pr\left[ \sum_{i=1}^{k} \mathbf{c}_i \cdot \mathbf{d}_i \leq O(1)k \log(1/\delta)p \right] \geq 1 - 2k \cdot e^{-O(\log(1/\delta))}$$

Thus with probability $1 - O(\delta)$,

$$\|\boldsymbol{U}^T\boldsymbol{S}^T\boldsymbol{S}\boldsymbol{W}\|_F^2 = \frac{1}{r^2}\sum_{i=1}^{k}\mathbf{c}_i \cdot \mathbf{d}_i$$

$$\leq \frac{1}{r^2}O(1)k\log(1/\delta)p$$

$$= \frac{1}{r^2}O(1)k\log(1/\delta)\|\boldsymbol{W}\|_F^2$$

Since $k < \log(1/\delta)$, $r = O(k + \log(1/\delta))$ in this case gives $\|\boldsymbol{U}^T\boldsymbol{S}^T\boldsymbol{S}\boldsymbol{W}\|_F \leq O(1)\|\boldsymbol{W}\|_F$ with probability $1 - \delta$.

Combining **Case I** and **Case II** allows us to conclude that for $r = O(k + \log(1/\delta))$, $\|\boldsymbol{U}^T\boldsymbol{S}^T\boldsymbol{S}\boldsymbol{W}\|_F \leq O(1)\|\boldsymbol{W}\|_F$ with probability $1 - \delta$.

$\square$

**Lemma B.1** (Upper Bound on Regression Error). *Given $\delta \in (0, \frac{1}{2})$, let $\boldsymbol{A}, \boldsymbol{B}$ be matrices that both have $n$ rows and $rank(\boldsymbol{A}) = k$. Let $\boldsymbol{S} \in \mathbb{R}^{n \times r}$ be a random matrix with i.i.d. $\mathcal{N}(0, \frac{1}{r})$ Gaussian random variables. Let $\widetilde{\boldsymbol{X}} = \arg\min_{\boldsymbol{X}} \|\boldsymbol{S}^T(\boldsymbol{A}\boldsymbol{X} - \boldsymbol{B})\|_F$ and $\boldsymbol{X}^* = \arg\min_{\boldsymbol{X}} \|\boldsymbol{A}\boldsymbol{X} - \boldsymbol{B}\|_F$. For $r = O(k + \log(1/\delta))$, the following holds with probability $1 - \delta$,*

$$\|\boldsymbol{A}\widetilde{\boldsymbol{X}} - \boldsymbol{B}\|_F \leq O(1)\|\boldsymbol{A}\boldsymbol{X}^* - \boldsymbol{B}\|_F$$

*Proof.* Consider an orthonormal basis $\boldsymbol{U}$ for the column span of $\boldsymbol{A}$. Let $\widetilde{\boldsymbol{Y}} = \arg\min_{\boldsymbol{Y}} \|\boldsymbol{S}\boldsymbol{U}\boldsymbol{Y} - \boldsymbol{S}\boldsymbol{B}\|_2$ and $\boldsymbol{Y}^* = \arg\min_{\boldsymbol{Y}} \|\boldsymbol{U}\boldsymbol{Y} - \boldsymbol{B}\|_2$. By the normal equations, the solutions to the two least squares problems are $\widetilde{\boldsymbol{Y}} = (\boldsymbol{S}\boldsymbol{U})^\dagger\boldsymbol{S}\boldsymbol{B}^1$ and $\boldsymbol{Y}^* = \boldsymbol{U}^T\boldsymbol{B}$.

We first show that $\|\boldsymbol{U}\widetilde{\boldsymbol{Y}} - \boldsymbol{B}\|_F \leq O(1)\|\boldsymbol{U}\boldsymbol{Y}^* - \boldsymbol{B}\|_F$.

$$\|\boldsymbol{U}\widetilde{\boldsymbol{Y}} - \boldsymbol{B}\|_F^2 = \|\boldsymbol{U}\boldsymbol{Y}^* - \boldsymbol{B}\|_F^2 + \|\boldsymbol{U}\widetilde{\boldsymbol{Y}} - \boldsymbol{U}\boldsymbol{Y}^*\|_F^2$$

$$= \|\boldsymbol{U}\boldsymbol{Y}^* - \boldsymbol{B}\|_F^2 + \|\widetilde{\boldsymbol{Y}} - \boldsymbol{Y}^*\|_F^2 \qquad \text{(Since } \boldsymbol{U} \text{ has orthonormal columns)}$$

$$= \|\boldsymbol{U}\boldsymbol{Y}^* - \boldsymbol{B}\|_F^2 + \|(\boldsymbol{S}\boldsymbol{U})^\dagger\boldsymbol{S}\boldsymbol{B} - \boldsymbol{U}^T\boldsymbol{B}\|_F^2$$

$$= \|\boldsymbol{U}\boldsymbol{Y}^* - \boldsymbol{B}\|_F^2 + \|(\boldsymbol{U}^T\boldsymbol{S}^T\boldsymbol{S}\boldsymbol{U})^{-1}\boldsymbol{U}^T\boldsymbol{S}^T\boldsymbol{S}\boldsymbol{B} - \boldsymbol{U}^T\boldsymbol{B}\|_F^2$$

Since $\boldsymbol{S}$ is a matrix with i.i.d. $\mathcal{N}(0, \frac{1}{r})$ Gaussian random variables, by **Fact** 3.3, for any vector $v \in \mathbb{R}^n$, with probability $1 - \delta$ and for some fixed constant $\epsilon_1 \in (0, 1)$, $\|\boldsymbol{S}\boldsymbol{U}v\|_2 = (1 \pm \epsilon_1)\|\boldsymbol{U}v\|_2$. This implies the singular values of $\boldsymbol{S}\boldsymbol{U}$ are in the range $[1 - \epsilon_1, 1 + \epsilon_1]$. Thus,

$$\|\boldsymbol{U}\widetilde{\boldsymbol{Y}} - \boldsymbol{B}\|_F^2 \leq \|\boldsymbol{U}\boldsymbol{Y}^* - \boldsymbol{B}\|_F^2 + O(1)\|(\boldsymbol{U}^T\boldsymbol{S}^T\boldsymbol{S}\boldsymbol{U})((\boldsymbol{U}^T\boldsymbol{S}^T\boldsymbol{S}\boldsymbol{U})^{-1}\boldsymbol{U}^T\boldsymbol{S}^T\boldsymbol{S}\boldsymbol{B} - \boldsymbol{U}^T\boldsymbol{B})\|_F^2$$

$$= \|\boldsymbol{U}\boldsymbol{Y}^* - \boldsymbol{B}\|_F^2 + O(1)\|\boldsymbol{U}^T\boldsymbol{S}^T\boldsymbol{S}\boldsymbol{B} - \boldsymbol{U}^T\boldsymbol{S}^T\boldsymbol{S}\boldsymbol{U}\boldsymbol{U}^T\boldsymbol{B}\|_F^2$$

$$= \|\boldsymbol{U}\boldsymbol{Y}^* - \boldsymbol{B}\|_F^2 + O(1)\|\boldsymbol{U}^T\boldsymbol{S}^T\boldsymbol{S}(\boldsymbol{B} - \boldsymbol{U}\boldsymbol{Y}^*)\|_F^2$$

Consider $p = \text{rank}(\boldsymbol{U}\boldsymbol{Y}^* - \boldsymbol{B})$. If $p = O(k)$, then $\text{rank}(\boldsymbol{B}) = O(k)$. For $r = O(k)$, we can use $\boldsymbol{S}$ to reconstruct $\boldsymbol{A}$ and $\boldsymbol{B}$. In this case, $\widetilde{\boldsymbol{X}} = \boldsymbol{X}^*$ and so $\|\boldsymbol{U}\widetilde{\boldsymbol{Y}} - \boldsymbol{B}\|_F \leq O(1)\|\boldsymbol{U}\boldsymbol{Y}^* - \boldsymbol{B}\|_F$. If $p = O(\log(1/\delta))$, then $\text{rank}(\boldsymbol{B}) = O(k + \log(1/\delta))$. For $r = O(k + \log(1/\delta))$, we can again use $\boldsymbol{S}$ to reconstruct $\boldsymbol{A}$ and $\boldsymbol{B}$ and get $\|\boldsymbol{U}\widetilde{\boldsymbol{Y}} - \boldsymbol{B}\|_F \leq O(1)\|\boldsymbol{U}\boldsymbol{Y}^* - \boldsymbol{B}\|_F$.

Now consider $p \geq \max(k, \log(1/\delta))$. First note that $\boldsymbol{B} - \boldsymbol{U}\boldsymbol{Y}^* = \boldsymbol{B} - \boldsymbol{U}\boldsymbol{U}^T\boldsymbol{B} = (\boldsymbol{I} - \boldsymbol{U}\boldsymbol{U}^T\boldsymbol{B})$, where $\boldsymbol{U}$ has orthonormal columns and thus, $\boldsymbol{U}\boldsymbol{U}^T$ is the projection matrix onto the column span $\text{col}(\boldsymbol{U})$ of $\boldsymbol{U}$. We have $(\boldsymbol{B} - \boldsymbol{U}\boldsymbol{Y}^*) \perp \text{col}(\boldsymbol{U})$. Second, we can w.l.o.g. assume that $\boldsymbol{U}\boldsymbol{Y}^* - \boldsymbol{B}$ has orthonormal columns; indeed, otherwise let $\boldsymbol{U}'\boldsymbol{R}' = \boldsymbol{B} - \boldsymbol{U}\boldsymbol{Y}^*$ be the QR decomposition where $\boldsymbol{U}'$ is an orthonormal basis for $\text{col}(\boldsymbol{B} - \boldsymbol{U}\boldsymbol{Y}^*)$. Then $\|\boldsymbol{U}^T\boldsymbol{S}^T\boldsymbol{S}(\boldsymbol{B} - \boldsymbol{U}\boldsymbol{Y}^*)\|_F^2 = \|\boldsymbol{U}^T\boldsymbol{S}^T\boldsymbol{S}\boldsymbol{U}'\boldsymbol{R}'\|_F^2 = \|\boldsymbol{U}^T\boldsymbol{S}^T\boldsymbol{S}\boldsymbol{U}'\|_F^2$.

Applying **Lemma** 3.4, with probability $1 - O(\delta)$,

$$\|\boldsymbol{U}\widetilde{\boldsymbol{Y}} - \boldsymbol{B}\|_F^2 \leq \|\boldsymbol{U}\boldsymbol{Y}^* - \boldsymbol{B}\|_F^2 + O(1)\|\boldsymbol{U}\boldsymbol{Y}^* - \boldsymbol{B}\|_F^2$$

---

$^1\dagger$ denotes the Moore-Penrose pseudoinverse

$$= O(1)\|\boldsymbol{U}\boldsymbol{Y}^* - \boldsymbol{B}\|_F^2$$

This concludes that $\|\boldsymbol{U}\widetilde{\boldsymbol{Y}} - \boldsymbol{B}\|_F \leq O(1)\|\boldsymbol{U}\boldsymbol{Y}^* - \boldsymbol{B}\|_F$.

Finally, consider the QR decomposition of $\boldsymbol{A} = \boldsymbol{U}\boldsymbol{R}$ where $\boldsymbol{U}$ is an orthonormal basis for the column span of $\boldsymbol{A}$ and $\boldsymbol{R}$ is an arbitrary matrix. Let $\widetilde{\boldsymbol{X}} = \arg\min_{\boldsymbol{X}} \|\boldsymbol{S}\boldsymbol{A}\boldsymbol{X} - \boldsymbol{S}\boldsymbol{B}\|_2$ and $\boldsymbol{X}^* = \|\boldsymbol{A}\boldsymbol{X} - \boldsymbol{B}\|_2$. Note that

$$\min_{\boldsymbol{X}} \|\boldsymbol{S}\boldsymbol{A}\boldsymbol{X} - \boldsymbol{S}\boldsymbol{B}\|_F = \min_{\boldsymbol{Y}} \|\boldsymbol{S}\boldsymbol{U}\boldsymbol{R}\boldsymbol{Y} - \boldsymbol{S}\boldsymbol{B}\|_F = \min_{\boldsymbol{Y}} \|\boldsymbol{S}\boldsymbol{U}\boldsymbol{Y} - \boldsymbol{S}\boldsymbol{B}\|_F$$

$$\min_{\boldsymbol{X}} \|\boldsymbol{A}\boldsymbol{X} - \boldsymbol{B}\|_F = \min_{\boldsymbol{Y}} \|\boldsymbol{U}\boldsymbol{R}\boldsymbol{Y} - \boldsymbol{B}\|_F = \min_{\boldsymbol{Y}} \|\boldsymbol{U}\boldsymbol{Y} - \boldsymbol{B}\|_F$$

Thus,

$$\|\boldsymbol{A}\widetilde{\boldsymbol{X}} - \boldsymbol{B}\|_F = \|\boldsymbol{U}\widetilde{\boldsymbol{Y}} - \boldsymbol{B}\|_F \leq O(1)\|\boldsymbol{U}\boldsymbol{Y}^* - \boldsymbol{B}\|_F = O(1)\|\boldsymbol{A}\boldsymbol{X}^* - \boldsymbol{B}\|_F$$

$\square$

The following Theorem and its proof follows **Theorem 4.7** of [9], except that: 1) to get a rank $k$ approximation to the matrix $\boldsymbol{A}$, the number of columns in the sketching matrices $\boldsymbol{S}$ and $\boldsymbol{R}$ was required to be $m = O(k\log(\frac{1}{\delta}))$ in **Theorem 4.7** of [9]; 2) $\boldsymbol{S}$ and $\boldsymbol{R}$ in **Theorem 4.7** of [9] are random sign matrices. By applying **Lemma** B.1, we show that this number $m$ can be reduced to $O(k + \log(\frac{1}{\delta}))$, and consider a specific application to PSD matrices.

**Theorem 3.5.** *Let $\boldsymbol{A} \in \mathbb{R}^{n \times n}$ be an arbitrary PSD matrix. Let $\boldsymbol{A}_k = \arg\min_{rank\text{-}kA_k} \|A - A_k\|_F$ be the optimal rank-k approximation to $\boldsymbol{A}$ in Frobenius norm. If $\boldsymbol{S} \in \mathbb{R}^{n \times m}$ and $\boldsymbol{R} \in \mathbb{R}^{n \times cm}$ are random matrices with i.i.d. $\mathcal{N}(0,1)$ entries for some fixed constant $c > 0$ with $m = O(k + \log(1/\delta))$, then with probability $1 - \delta$, the matrix $\widetilde{\boldsymbol{A}} = (\boldsymbol{A}\boldsymbol{R})(\boldsymbol{S}^T\boldsymbol{A}\boldsymbol{R})^\dagger(\boldsymbol{A}\boldsymbol{S})^T$ satisfies*

$$\|\boldsymbol{A} - \widetilde{\boldsymbol{A}}\|_F \leq O(1)\|\boldsymbol{A} - \boldsymbol{A}_k\|_F$$

*Proof.* First, we consider $\boldsymbol{S}$ to be a random matrix with i.i.d. $\mathcal{N}(0, \frac{1}{m})$ entries and $\boldsymbol{R}$ to be a random matrix with i.i.d. $\mathcal{N}(0, \frac{1}{cm})$ entries.

Consider $\widetilde{\boldsymbol{X}} = \arg\min_{\boldsymbol{X}} \|\boldsymbol{S}^T\boldsymbol{A}\boldsymbol{R}\boldsymbol{X} - \boldsymbol{S}^T\boldsymbol{A}\|_F = (\boldsymbol{S}^T\boldsymbol{A}\boldsymbol{R})^\dagger\boldsymbol{S}^T\boldsymbol{A}$
and $\boldsymbol{X}^* = \arg\min_{\boldsymbol{X}} \|\boldsymbol{A}\boldsymbol{R}\boldsymbol{X} - \boldsymbol{A}\|_F$. By **Lemma** B.1, with probability $1 - \delta$,

$$\|\boldsymbol{A}\boldsymbol{R}\widetilde{\boldsymbol{X}} - \boldsymbol{A}\|_F \leq O(1)\|\boldsymbol{A}\boldsymbol{R}\boldsymbol{X}^* - \boldsymbol{A}\|_F$$

Now let $\boldsymbol{A}_k = \arg\min_{\text{rank k } A_k} \|\boldsymbol{A} - \boldsymbol{A}_k\|_F$ be the optimal rank-$k$ approximation to $\boldsymbol{A}$.

Consider $\boldsymbol{X}_{opt} = \arg\min_{\boldsymbol{X}} \|\boldsymbol{X}\boldsymbol{A}_k - \boldsymbol{A}\|_F$ and $\boldsymbol{X}' = \arg\min_{\boldsymbol{X}} \|\boldsymbol{X}\boldsymbol{A}_k\boldsymbol{R} - \boldsymbol{A}\boldsymbol{R}\|_F = (\boldsymbol{A}\boldsymbol{R})(\boldsymbol{A}_k\boldsymbol{R})^\dagger$.

By **Lemma** B.1 again, with probability $1 - \delta$,

$$\|\boldsymbol{X}'\boldsymbol{A}_k - \boldsymbol{A}\|_F = \|(\boldsymbol{A}\boldsymbol{R})(\boldsymbol{A}_k\boldsymbol{R})^\dagger\boldsymbol{A}_k - \boldsymbol{A}\|_F$$
$$\leq O(1)\|\boldsymbol{X}_{opt}\boldsymbol{A}_k - \boldsymbol{A}\|_F = O(1)\|\boldsymbol{A} - \boldsymbol{A}_k\|_F$$

This implies a good rank-$k$ approximation exists in the column span of $\boldsymbol{A}\boldsymbol{R}$. We now have with probability $1 - \delta$,

$$\|\boldsymbol{A}\boldsymbol{R}\boldsymbol{X}^* - \boldsymbol{A}\|_F \leq \|(\boldsymbol{A}\boldsymbol{R})(\boldsymbol{A}_k\boldsymbol{R})^\dagger\boldsymbol{A}_k - \boldsymbol{A}\|_F \leq O(1)\|\boldsymbol{A} - \boldsymbol{A}_k\|_F$$

Thus by a union bound, with probability $1 - 2\delta$,

$$\|\boldsymbol{A}\boldsymbol{R}(\boldsymbol{S}^T\boldsymbol{A}\boldsymbol{R})^\dagger\boldsymbol{S}^T\boldsymbol{A} - \boldsymbol{A}\|_F = \|\boldsymbol{A}\boldsymbol{R}\widetilde{\boldsymbol{X}} - \boldsymbol{A}\|_F$$
$$\leq O(1)\|\boldsymbol{A}\boldsymbol{R}\boldsymbol{X}^* - \boldsymbol{A}\|_F$$
$$\leq O(1)\|\boldsymbol{A} - \boldsymbol{A}_k\|_F$$

Since we consider PSD $\boldsymbol{A}$, $\boldsymbol{S}^T\boldsymbol{A} = (\boldsymbol{A}\boldsymbol{S})^T$. Let $\widetilde{\boldsymbol{A}} = (\boldsymbol{A}\boldsymbol{R})(\boldsymbol{S}^T\boldsymbol{A}\boldsymbol{R})^\dagger(\boldsymbol{A}\boldsymbol{S})^T$, it follows that with probability $1 - 2\delta$,

$$\|\boldsymbol{A} - \widetilde{\boldsymbol{A}}\|_F \leq O(1)\|\boldsymbol{A} - \boldsymbol{A}_k\|_F$$

Let $\boldsymbol{S}' = \sqrt{m}\boldsymbol{S}$ and $\boldsymbol{R}' = \sqrt{cm}\boldsymbol{R}$ so that both $\boldsymbol{S}'$ and $\boldsymbol{R}'$ have i.i.d. $\mathcal{N}(0,1)$ entries. Notice that $(\boldsymbol{A}\boldsymbol{R}')(\boldsymbol{S}'^T\boldsymbol{A}\boldsymbol{R}')^\dagger(\boldsymbol{A}\boldsymbol{S}')^T = (\boldsymbol{A}\boldsymbol{R})(\boldsymbol{S}^T\boldsymbol{A}\boldsymbol{R})^\dagger(\boldsymbol{A}\boldsymbol{S})^T$. Thus $\boldsymbol{S}, \boldsymbol{R}$ can be chosen to both be random matrices with i.i.d. $\mathcal{N}(0,1)$ entries. The theorem follows after adjusting $\delta$ by a constant factor. $\square$

**Theorem 3.2 (Theorem 4 of [7]).** *Let $\boldsymbol{A} \in \mathbb{R}^{d \times d}$ be PSD, $\delta \in (0, \frac{1}{2})$, $l \in \mathbb{N}, k \in \mathbb{N}$. Let $\widetilde{\boldsymbol{A}}$ and $\boldsymbol{\Delta}$ be any matrices with $\mathrm{tr}(\boldsymbol{A}) = \mathrm{tr}(\widetilde{\boldsymbol{A}}) + \mathrm{tr}(\boldsymbol{\Delta})$ and $\|\boldsymbol{\Delta}\|_F \leq O(1)\|\boldsymbol{A} - \boldsymbol{A}_k\|_F$ where $\boldsymbol{A}_k = \arg\min_{rank\ k\ \boldsymbol{A}_k} \|\boldsymbol{A} - \boldsymbol{A}_k\|_F$. Let $H_l(\boldsymbol{M})$ denote Hutchinson's trace estimator with $l$ queries on matrix $\boldsymbol{M}$. For fixed constants $c, C$, if $l \geq c\log(\frac{1}{\delta})$, then with probability $1 - \delta$, $Z = \mathrm{tr}(\widetilde{\boldsymbol{A}}) + H_l(\boldsymbol{\Delta})$,*

$$|Z - \mathrm{tr}(\boldsymbol{A})| \leq C\sqrt{\frac{\log(1/\delta)}{kl}} \cdot \mathrm{tr}(\boldsymbol{A})$$

**Theorem 3.1.** *Let $\boldsymbol{A}$ be a PSD matrix. If `NA-Hutch++` is implemented with*

$$m = O\left(\frac{\sqrt{\log(1/\delta)}}{\epsilon} + \log(1/\delta)\right)$$

*matrix-vector multiplication queries, then with probability $1 - \delta$, the output of `NA-Hutch++`, $t$, satisfies $(1 - \epsilon)\mathrm{tr}(\boldsymbol{A}) \leq t \leq (1 + \epsilon)\mathrm{tr}(\boldsymbol{A})$.*

*Proof.* Set $k = l = O(\frac{\sqrt{\log(1/\delta)}}{\epsilon})$.

Consider $\widetilde{\boldsymbol{A}} = (\boldsymbol{AR})(\boldsymbol{S}^T\boldsymbol{AR})^\dagger(\boldsymbol{AS})^T$, where $\boldsymbol{S} \in \mathbb{R}^{n \times s}, \boldsymbol{R} \in \mathbb{R}^{n \times r}$ are both random matrices with i.i.d. $\mathcal{N}(0, 1)$ entries, and $\boldsymbol{\Delta} = \boldsymbol{A} - \widetilde{\boldsymbol{A}}$.

By **Theorem** 3.5, for $s = r = O(k + \log(1/\delta)) = O(\frac{\sqrt{\log(1/\delta)}}{\epsilon} + \log(1/\delta))$, with probability $1 - \delta$,

$$\|\boldsymbol{\Delta}\|_F \leq O(1) \cdot \|\boldsymbol{A} - \boldsymbol{A}_k\|_F$$

Thus for the output of `NA-Hutch++`, $t$, by **Theorem** 3.2 and a union bound, with probability $1 - 2\delta$,

$$|t - \mathrm{tr}(\boldsymbol{A})| \leq \epsilon \cdot \mathrm{tr}(\boldsymbol{A})$$

The total number of non-adaptive queries `NA-Hutch++` needs is

$$m = s + r + l = O\left(\frac{\sqrt{\log(1/\delta)}}{\epsilon} + \log(1/\delta)\right).$$

$\square$

# C  Lower Bounds

In this section, we show that a query complexity of $O\left(\frac{\sqrt{\log(1/\delta)}}{\epsilon} + \log(1/\delta)\right)$ is tight for any non-adaptive trace estimation algorithm, up to a $O(\log\log(1/\delta))$ factor, stated in **Theorem** 4.1. The analysis considers two separate cases: for small $\epsilon$, we show the term $O\left(\frac{\sqrt{\log(1/\delta)}}{\epsilon}\right)$ is tight in Section C.1, and for any $\epsilon$, we show the term $O(\log(1/\delta))$ is tight up to a $O(\log\log(1/\delta))$ factor in Section C.2. When combined, these two lower bounds handle arbitrary $\epsilon$, since the latter lower bound dominates precisely when the former lower bound does not apply.

Our hard distribution consists of shifted Wigner matrices and exploits the symmetry and concentration properties of the Gaussian ensemble.

**Theorem 4.1** (Lower Bound for Non-Adaptive Queries). *Let $\epsilon \in (0, 1)$. Any algorithm that accesses a real PSD matrix $\boldsymbol{A}$ through matrix-vector multiplication queries $\boldsymbol{A}\mathbf{q}_1, \boldsymbol{A}\mathbf{q}_2, \ldots, \boldsymbol{A}\mathbf{q}_m$, where $\mathbf{q}_1, \ldots, \mathbf{q}_m$ are real-valued, non-adaptively chosen vectors, requires*

$$m = \Omega\left(\frac{\sqrt{\log(1/\delta)}}{\epsilon} + \frac{\log(1/\delta)}{\log\log(1/\delta)}\right)$$

*queries to output an estimate $t$ such that with probability at least $1 - \delta$, $(1-\epsilon)\mathrm{tr}(\boldsymbol{A}) \leq t \leq (1+\epsilon)\mathrm{tr}(\boldsymbol{A})$.*

*Proof of Theorem 4.1.* For small $\epsilon = O(1/\sqrt{\log(1/\delta)})$, note that the first term $\frac{\sqrt{\log(1/\delta)}}{\epsilon}$ dominates. **Theorem** 4.2 (see Section C.1) shows any algorithm needs $\Omega\left(\frac{\sqrt{\log(1/\delta)}}{\epsilon}\right)$ non-adaptive queries in this case.

For $\epsilon > 1/\sqrt{\log(1/\delta)}$, note that the second term $\log(1/\delta)$ dominates. **Theorem** 4.3 (see Section C.2) shows any algorithm needs $\Omega(\frac{\log(1/\delta)}{\log\log(1/\delta)})$ non-adaptive queries for any $\epsilon \in (0, 1)$.

The two cases combined imply an $\Omega\left(\frac{\sqrt{\log(1/\delta)}}{\epsilon} + \frac{\log(1/\delta)}{\log\log(1/\delta)}\right)$ lower bound. $\qquad\square$

## C.1  Case 1: Lower Bound for Small $\epsilon$

Suppose that we draw a matrix $\boldsymbol{G} \in \mathbb{R}^{n \times n}$ from the Gaussian distribution and try to learn the entries of the matrix via matrix-vector queries. After a few queries, it turns out that the conditional distribution of the remaining matrix is also Gaussian-distributed, no matter how the queries are chosen. This nice property allows concise reasoning for lower bounding the remaining uncertainty of the matrix, even after seeing a few query results.

**Lemma C.1.** *(Conditional Distribution [Lemma 3.4 of [10]]) Let $\boldsymbol{G} \sim \mathcal{N}(n)$ be as in Definition A.1 and suppose our matrix is $\boldsymbol{W} = (\boldsymbol{G} + \boldsymbol{G}^\top)/2$. Suppose we have any sequence of vector queries, $\boldsymbol{v}_1, ..., \boldsymbol{v}_T$, along with responses $\boldsymbol{w}_i = \boldsymbol{W}\boldsymbol{v}_i$. Then, conditioned on our observations, there exists a rotation matrix $\boldsymbol{V}$, independent of $\boldsymbol{w}_i$, such that*

$$\boldsymbol{V}\boldsymbol{W}\boldsymbol{V}^\top = \begin{bmatrix} Y_1 & Y_2^\top \\ Y_2 & \widetilde{\boldsymbol{W}} \end{bmatrix}$$

*where $Y_1, Y_2$ are deterministic and $\widetilde{\boldsymbol{W}} = (\widetilde{\boldsymbol{G}} + \widetilde{\boldsymbol{G}}^\top)/2$, where $\widetilde{\boldsymbol{G}} \sim \mathcal{N}(n - T)$.*

**Theorem 4.2** (Lower Bound for Small $\epsilon$). *For any PSD matrix $\boldsymbol{A}$ and all $\epsilon = O(1/\sqrt{\log(1/\delta)})$, any algorithm that succeeds with probability at least $1 - \delta$ in outputting an estimate $t$ such that $(1 - \epsilon)\mathrm{tr}(\boldsymbol{A}) \leq t \leq (1 + \epsilon)\mathrm{tr}(\boldsymbol{A})$, requires*

$$m = \Omega(\sqrt{\log(1/\delta)}/\epsilon)$$

*matrix-vector queries.*

*Proof.* By standard minimax arguments, it suffices to construct a hard distribution for any deterministic algorithm.

Consider $\boldsymbol{G} \sim \mathcal{N}(n)$ for $n = \Omega(\log(1/\delta))$. From concentration of the singular values of large Gaussian matrices (Lemma A.2), with probability at least $1 - \delta/10$ we have $\|\boldsymbol{G}\|_{op} \leq C\sqrt{n}$ for some absolute constant $C$.

Therefore, consider the family of matrices $\boldsymbol{W} = \boldsymbol{I} + \frac{1}{2C\sqrt{n}}(\boldsymbol{G} + \boldsymbol{G}^\top)$. From our bound on $\|\boldsymbol{G}\|_{op}$, with probability at least $1 - \delta/10$, $\boldsymbol{W}$ is positive semi-definite and symmetric. Furthermore, since $\mathrm{tr}(\boldsymbol{G}) \sim N(0, n)$, we see that $\mathrm{tr}(\boldsymbol{W}) \leq 2n$ with probability at least $1 - \delta/10$.

We set the multiplicative error to $\epsilon = \frac{\sqrt{\log(1/\delta)}}{n}$ and it suffices to show that if we see only $n/2$ queries, we can compute $\mathrm{tr}(\boldsymbol{W})$ up to additive error at best $c\sqrt{\log(1/\delta)}$ with probability at least $1 - \delta$, for some $c = \Omega(1)$. By **Lemma** C.1, we see that conditioned on the queries, our matrix $\boldsymbol{W}$ can be decomposed into a determined part and a Gaussian submatrix $\widetilde{\boldsymbol{W}} = \frac{1}{2C\sqrt{n}}(\widetilde{\boldsymbol{G}} + \widetilde{\boldsymbol{G}}^\top)$, where $\widetilde{\boldsymbol{G}} \sim \mathcal{N}(n/2)$.

Therefore, our conditional distribution of the trace of $\boldsymbol{W}$ is, up to a deterministic shift, the same as the distribution of $\widetilde{\boldsymbol{W}}$, which is simply a Gaussian with variance $1/C^2$. Since we must determine a Gaussian of constant variance up to an additive error of $c\sqrt{\log(1/\delta)}$ with probability at least $1 - \delta$, we conclude that $c = \Omega(1)$. $\qquad\square$

## C.2   Case 2: Lower Bound for Every $\epsilon$

We give a general $\Omega(\frac{\log(1/\delta)}{\log\log(1/\delta)})$ lower bound, that holds for every $\epsilon \in (0, 1)$, on the query complexity for non-adaptive trace estimation algorithms stated in **Theorem** 4.3. The proof of **Theorem** 4.3 is via a reduction to a distribution testing problem in **Problem** 4.4, whose hardness (in terms of query complexity) is shown in **Lemma** 4.5.

**Theorem 4.3** (Lower Bound on Non-adaptive Queries for PSD Matrices). *Let $\epsilon \in (0, 1)$. Any algorithm that accesses a real, PSD matrix $\boldsymbol{A}$ through matrix-vector queries $\boldsymbol{A}\mathbf{q}_1, \boldsymbol{A}\mathbf{q}_2, \ldots, \boldsymbol{A}\mathbf{q}_m$, where $\mathbf{q}_1, \ldots, \mathbf{q}_m$ are real-valued non-adaptively chosen vectors, requires*

$$m = \Omega\left(\frac{\log(1/\delta)}{\log\log(1/\delta)}\right)$$

*to output an estimate $t$ such that with probability at least $1 - \delta$, $(1 - \epsilon)\mathrm{tr}(\boldsymbol{A}) \leq t \leq (1 + \epsilon)\mathrm{tr}(\boldsymbol{A})$.*

*Proof.* The proof is via reduction to a distribution testing problem stated in **Problem** 4.4. Given a real, PSD input matrix $\boldsymbol{A}$, let $\mathcal{A}$ be an algorithm that uses $m$ non-adaptive matrix-vector queries and outputs a trace estimation $t$ of $\boldsymbol{A}$ such that for some $\epsilon \in (0, 1)$, with probability at least $1 - \delta$, $(1 - \epsilon)\mathrm{tr}(\boldsymbol{A}) \leq t \leq (1 + \epsilon)\mathrm{tr}(\boldsymbol{A})$.

Consider $n = \log(1/\delta)$. Let $Z_i, \forall i \in [n]$ be the $i$-th diagonal entry of $\boldsymbol{W} \sim \mathcal{W}(n) = \boldsymbol{G} + \boldsymbol{G}^T$ as in Definition A.1. Note that $\boldsymbol{G}$ has i.i.d. $\mathcal{N}(0, 1)$ entries, and that the diagonal of $\boldsymbol{G}$ and $\boldsymbol{G}^T$ are the same. This implies $Z_i \sim \mathcal{N}(0, 4)$.

Since the $Z_i$ are i.i.d.,

$$\mathrm{tr}(\boldsymbol{W}) = \sum_{i=1}^{n} Z_i \sim \mathcal{N}(0, 4n) = \mathcal{N}(0, 4\log(1/\delta))$$

By **Fact** A.3,

$$\Pr[\mathrm{tr}(\boldsymbol{W}) \geq 2\sqrt{2}\log(1/\delta)] \leq \delta$$
$$\Pr[\mathrm{tr}(\boldsymbol{W}) \leq -2\sqrt{2}\log(1/\delta)] \leq \delta$$

For a unit vector $\frac{\mathbf{g}}{\|\mathbf{g}\|_2} \in \mathbb{R}^n$,

$$\mathrm{tr}\left(\frac{\mathbf{g}}{\|\mathbf{g}\|_2}\frac{\mathbf{g}^T}{\|\mathbf{g}\|_2}\right) = \left\|\frac{\mathbf{g}}{\|\mathbf{g}\|_2}\right\|_2^2 = 1$$

Let $\boldsymbol{B}$ be the random matrix generated from distribution $\mathcal{P}$ or $\mathcal{Q}$ in **Problem** 4.4. First, we claim that with probability at least $1 - 4\delta$, $\boldsymbol{B}$ is a PSD matrix. Note that $C\log^{3/2}(\frac{1}{\delta}) \cdot \frac{1}{\|\mathbf{g}\|_2^2}\mathbf{g}\mathbf{g}^T$ is PSD. Thus it suffices to show $\boldsymbol{W} + 6\sqrt{\log(\frac{1}{\delta})}\boldsymbol{I}$ is PSD with high probability.

By **Lemma** A.2, with probability $1 - 2\delta$,

$$\|\boldsymbol{G}\|_{op} \leq 3\sqrt{\log(1/\delta)}$$

By the triangle inequality and a union bound, with probability $1 - 4\delta$,

$$\|\boldsymbol{W}\|_{op} = \|\boldsymbol{G} + \boldsymbol{G}^T\|_{op} \leq 6\sqrt{\log(1/\delta)}$$

This implies $\boldsymbol{W} + 6\sqrt{\log(\frac{1}{\delta})}\boldsymbol{I}$ is PSD with probability $1 - 4\delta$.

If $\boldsymbol{B} \sim \mathcal{P}$, with probability at least $1 - \delta$,

$$
\begin{aligned}
\text{tr}(\boldsymbol{B}) &= C\log^{3/2}(1/\delta) + \text{tr}(\boldsymbol{W}) + 6\log^{3/2}(1/\delta) \\
&\geq (C + 6)\log^{3/2}(1/\delta) - 2\sqrt{2}\log(1/\delta)
\end{aligned}
$$

If $\boldsymbol{B} \sim \mathcal{Q}$, with probability at least $1 - \delta$,

$$\text{tr}(\boldsymbol{B}) = \text{tr}(\boldsymbol{W}) + 6\log^{3/2}(\log(1/\delta)) \leq 2\sqrt{2}\log(1/\delta) + 6\log^{3/2}(1/\delta)$$

Consider the trace estimation algorithm $\mathcal{A}$ and let the output $t = \mathcal{A}(\boldsymbol{B})$. Consider the constant $C > \frac{10(1+\epsilon)}{1-\epsilon} - 6$. If $\boldsymbol{B} \sim \mathcal{P}$, with probability at least $1 - 2\delta$,

$$
\begin{aligned}
t &\geq (1 - \epsilon)\text{tr}(\boldsymbol{B}) \\
&\geq (1 - \epsilon)\left((C + 6)\log^{3/2}(1/\delta) - 2\sqrt{2}\log(1/\delta)\right) \\
&> 6(1 + \epsilon)\log^{3/2}(1/\delta)
\end{aligned}
$$

If $\boldsymbol{B} \sim \mathcal{Q}$, with probability at least $1 - 2\delta$,

$$
\begin{aligned}
t &\leq (1 + \epsilon)\text{tr}(\boldsymbol{B}) \\
&\leq (1 + \epsilon)\left(6\log^{3/2}(1/\delta) + 2\sqrt{2}\log(1/\delta)\right) \\
&< 6(1 + \epsilon)\log^{3/2}(1/\delta)
\end{aligned}
$$

In the worst case, if any of the instances generated from $\mathcal{P}$ or $\mathcal{Q}$ is non-PSD, our algorithm $\mathcal{A}$ fails. Thus $\mathcal{A}$ determines which distribution $\boldsymbol{B}$ comes from with probability at least $1 - 6\delta$. By **Lemma** 4.5, this requires the number of matrix-vector queries $\mathcal{A}$ uses to be $m = \Omega(\frac{\log(1/\delta)}{\log\log(1/\delta)})$.

$\square$

**Problem 4.4** (Hard PSD Matrix Distribution Test). *Given $\delta \in (0, \frac{1}{2})$, set $n = \log(1/\delta)$. Choose $\mathbf{g} \in \mathbb{R}^n$ to be an independent random vector with i.i.d. $\mathcal{N}(0,1)$ entries. Consider two distributions:*

- *Distribution $\mathcal{P}$ on matrices $\left\{C\log^{3/2}(\frac{1}{\delta}) \cdot \frac{1}{\|\mathbf{g}\|_2^2}\mathbf{g}\mathbf{g}^T + \boldsymbol{W} + 6\sqrt{\log(\frac{1}{\delta})}\boldsymbol{I}\right\}$, for some fixed constant $C > 1$.*

- *Distribution $\mathcal{Q}$ on matrices $\left\{\boldsymbol{W} + 6\sqrt{\log(\frac{1}{\delta})}\boldsymbol{I}\right\}$.*

*where $\boldsymbol{W} \sim \mathcal{W}(n) = \boldsymbol{G} + \boldsymbol{G}^T$ as in Definition A.1. Let $\boldsymbol{A}$ be a random matrix drawn from either $\mathcal{P}$ or $\mathcal{Q}$ with equal probability. Consider any algorithm which, for a fixed query matrix $\boldsymbol{Q} \in \mathbb{R}^{n \times q}$, observes $\boldsymbol{A}\boldsymbol{Q}$, and guesses if $\boldsymbol{A} \sim \mathcal{P}$ or $\boldsymbol{A} \sim \mathcal{Q}$ with success probability at least $1 - \delta$.*

**Lemma 4.5** (Hardness of Problem 4.4). *Given $\delta \in (0, \frac{1}{2})$. Consider a non-adaptively chosen query matrix $\boldsymbol{Q} \in \mathbb{R}^{n \times q}$ on input $\boldsymbol{A} \in \mathbb{R}^{n \times n}$, as in **Problem** 4.4, where $n = \log(1/\delta)$. If $q = o(\frac{\log(1/\delta)}{\log\log(1/\delta)})$, no algorithm can solve **Problem** 4.4 with success probability $1 - \delta$.*

*Proof.* We claim that without loss of generality, we only need to consider $\boldsymbol{Q}$ to be the first $q$ standard basis vectors, i.e., $\boldsymbol{Q} = \boldsymbol{E}_q = [\mathbf{e}_1, \mathbf{e}_2, \ldots, \mathbf{e}_q]$. First note that we only need to consider query matrix $\boldsymbol{Q}$ with orthonormal columns, since for general $\boldsymbol{Q}$, letting $\boldsymbol{Q} = \boldsymbol{U}\boldsymbol{R}$ be the QR decomposition of $\boldsymbol{Q}$, we can reconstruct $\boldsymbol{A}\boldsymbol{Q}$ from $(\boldsymbol{A}\boldsymbol{U})\boldsymbol{R}$. Next, let $\bar{\boldsymbol{Q}} \in \mathbf{R}^{n \times (n-q)}$ be the orthonormal basis for null$(\boldsymbol{Q})$. Define an orthornomal matrix $\boldsymbol{R} = [\boldsymbol{Q}, \bar{\boldsymbol{Q}}] \in \mathbf{R}^{n \times n}$. By **Fact** A.2, $\boldsymbol{W}\boldsymbol{E}_q$ has the same distribution as $\boldsymbol{W}\boldsymbol{R}\boldsymbol{E}_q = \boldsymbol{W}\boldsymbol{Q}$. Similarly, $(C\log(\frac{1}{\delta}) \cdot \frac{1}{\|\mathbf{g}\|_2^2}\mathbf{g}\mathbf{g}^T + \boldsymbol{W})\boldsymbol{E}_q$ has the same distribution as

$(C \log(\frac{1}{\delta}) \cdot \frac{1}{\|\mathbf{g}\|_2^2} \mathbf{g}\mathbf{g}^T + \mathbf{W})\mathbf{Q}$. Therefore, we only need to consider the case when the queries are the first $q$ standard basis vectors.

Consider the two possible observed distributions from **Problem** 4.4: 1) distribution $\mathcal{P}'$, which has $(C \log(\frac{1}{\delta}) \cdot \frac{1}{\|\mathbf{g}\|_2^2} \mathbf{g}\mathbf{g}^T + \mathbf{W} + 2\sqrt{\log(1/\delta)}\mathbf{I})\mathbf{Q}$ for fixed constant $C > 1$, and 2) distribution $\mathcal{Q}'$ which has $(\mathbf{W} + 2\sqrt{\log(1/\delta)}\mathbf{I})\mathbf{Q}$.

We argue that if the number $q$ of queries is too small, then the total variation distance between $\mathcal{P}'$ and $\mathcal{Q}'$, conditioned on an event $\mathcal{E}$ with probability at least $\delta$, is upper bounded by a small constant. This will imply that no algorithm can succeed with probability at least $1 - \delta$. We upper bound the total variation distance between $\mathcal{P}'$ and $\mathcal{Q}'$ via the Kullback–Leibler (KL) divergence between $\mathcal{P}'$ and $\mathcal{Q}'$ and then apply Pinsker's inequality.

Consider the following event on over the randomness of $\mathbf{g}$: $\mathcal{E} = \left\{\mathbf{g} : \frac{1}{\|\mathbf{g}\|^2} \|\mathbf{g}^T\mathbf{Q}\|^2 \le \frac{1}{50C^2 n^3}\right\}$. Note that $\mathbf{g}^T\mathbf{Q} = [\langle \mathbf{g}, \mathbf{e}_1 \rangle, \langle \mathbf{g}, \mathbf{e}_2 \rangle, \ldots, \langle \mathbf{g}, \mathbf{e}_q \rangle] = [\mathbf{g}_1, \mathbf{g}_2, \ldots, \mathbf{g}_q]$, i.e., the first $q$ coordinates of $\mathbf{g}$. First, we show that $\Pr[\mathcal{E}] = \Omega(\delta)$.

Since $\mathbf{g}_i \sim \mathcal{N}(0, 1)$, by **Fact** A.4, for the $i$-th entry of $\mathbf{g}^T\mathbf{Q}$, $\forall i \in [q]$,

$$\Pr[|\mathbf{g}_i| \le \frac{1}{10C \cdot n\sqrt{q}}] = \Omega(\frac{1}{n\sqrt{q}})$$

which implies for a single entry,

$$\Pr[\mathbf{g}_i^2 \le \frac{1}{100C^2 \cdot n^2 q}] = \Omega(\frac{1}{n\sqrt{q}})$$

Since all $q$ queries are independent, for all entries $i \in [q]$,

$$\Pr[\|\mathbf{g}^T\mathbf{Q}\|_2^2 \le \frac{1}{100C^2 \cdot n^2}] = \Omega((\frac{1}{n\sqrt{q}})^q) = \Omega(\exp(-\frac{q}{2}\ln(n^2 q)))$$

Consider the following conditional probability,

$$\Pr\left[\|\mathbf{g}^T\mathbf{Q}\|_2^2 \le \frac{1}{100C^2 \cdot n^2} \wedge \|\mathbf{g}\|_2^2 \ge \frac{n}{2}\right]$$
$$= \Pr\left[\|\mathbf{g}\|_2^2 \ge \frac{n}{2} \;\middle|\; \|\mathbf{g}^T\mathbf{Q}\|_2^2 \le \frac{1}{100C^2 \cdot n^2}\right] \cdot \Pr\left[\|\mathbf{g}^T\mathbf{Q}\|_2^2 \le \frac{1}{100C^2 \cdot n^2}\right]$$

Assume $q < \frac{n}{2}$ and let $\mathbf{g}_{(q+1):n}$ denote the $q+1$-th to the $n$-th entry of $\mathbf{g}$. Note that all entries of $\mathbf{g}$ are independent and $\|\mathbf{g}_{(q+1):n}\|_2^2 \sim \chi^2(d)$ with degree $d > \frac{n}{2}$. By **Fact** A.1, since $\|\mathbf{g}\|_2^2 \ge \|\mathbf{g}_{(q+1):n}\|_2^2$,

$$\Pr\left[\|\mathbf{g}\|_2^2 \ge \frac{n}{2} \;\middle|\; \|\mathbf{g}^T\mathbf{Q}\|_2^2 \le \frac{1}{100C^2 \cdot n^2}\right] = \Omega(1)$$

Thus,

$$\Pr\left[\frac{1}{\|\mathbf{g}\|_2^2}\|\mathbf{g}^T\mathbf{Q}\|_2^2 \le \frac{1}{50C^2 n^3}\right] \ge \Pr\left[\|\mathbf{g}^T\mathbf{Q}\|_2^2 \le \frac{1}{100C^2 \cdot n^2} \wedge \|\mathbf{g}\|_2^2 \ge \frac{n}{2}\right]$$
$$\ge \Omega(1) \cdot \Omega\left(\exp(-\frac{q}{2}\ln(n^2 q))\right)$$

Assume we only have a small number $q = o(\frac{\log(1/\delta)}{\log\log(1/\delta)})$ of queries. Then,

$$\Pr[\mathcal{E}] = \Pr\left[\frac{1}{\|\mathbf{g}\|_2^2}\|\mathbf{g}^T\mathbf{Q}\|_2^2 \le \frac{1}{50C^2 \cdot n^3}\right] \ge 10\delta \tag{1}$$

Note that $n = \log(1/\delta)$, and so

$$\Pr[\mathcal{E}] = \Pr[C^2 \log^3(\frac{1}{\delta})\frac{\|\mathbf{g}^T\mathbf{Q}\|_2^2}{\|\mathbf{g}\|_2^2} \le \frac{1}{50}] \ge 10\delta$$

Next, note that it suffices to show that the probability of success conditioned on $\mathcal{E}$ is less than $1/3$. This implies our result since $\mathcal{E}$ occurs with probability at least $10\delta$, implying that our probability of failure is indeed $\Omega(\delta)$. Therefore, we focus on showing that the probability of success conditioned on $\mathbf{g} \in \mathcal{E}$ is small via standard information theoretic arguments with KL divergence bounds.

Conditioning on event $\mathcal{E}$, we now upper bound the KL divergence between $\mathcal{P}'$ and $\mathcal{Q}'$ conditioned on a fixed $\mathbf{g} \in \mathcal{E}$. Since both distributions come from symmetric matrices, we remove the redundant random variables from observed random matrices from $\mathcal{P}', \mathcal{Q}'$ and consider only the lower triangular portion, so that both have dimensions $l = n + (n-1) + \cdots + (n - (q-1))$. Note that these redundant random variables in the upper triangular portion can be removed without increasing the KL divergence, since they are perfectly correlated with its counterpart variable in the lower triangular region, which we show as follows:

Consider two lists $L_{\mathcal{P}'}, L_{\mathcal{Q}'}$ of $l$ random variables, corresponding to a vectorization of the observed lower triangular part of the random matrices from $\mathcal{P}'$ and $\mathcal{Q}'$. Consider also a function $f$, which duplicates parts of the random variables in $L_{\mathcal{P}'}$ and $L_{\mathcal{Q}'}$, such that $f(L_{\mathcal{P}'})$ and $f(L_{\mathcal{Q}'})$ reconstruct the original observed matrix of size $n \times q$ from $\mathcal{P}'$ and $\mathcal{Q}'$, respectively. Then, by the data processing inequality of KL divergence from **Fact** A.7,

$$\mathcal{D}_{KL}(\mathcal{P}' \parallel \mathcal{Q}') = \mathcal{D}_{KL}(f(L_{\mathcal{P}'}) \parallel f(L_{\mathcal{Q}'})) \leq \mathcal{D}_{KL}(L_{\mathcal{P}'} \parallel L_{\mathcal{Q}'})$$

From now on, we assume that $\mathcal{P}', \mathcal{Q}'$ are lower triangular. The KL divergence between $\mathcal{P}'|\mathbf{g}$ and $\mathcal{Q}'|\mathbf{g}$ considering the lower triangular part can be calculated since they are both multivariate Gaussians with the same covariance matrix (of rank $l$). The KL divergence thus only depends on the difference between the mean $\Delta\mu$ of the two multivariate Gaussians (see Fact A.5), which is the lower triangular part contained in $C\log^{3/2}(\frac{1}{\delta})\frac{\mathbf{g}\mathbf{g}^T}{\|\mathbf{g}\|_2^2}\mathbf{Q}$. Furthermore, since all redundant variables are removed, the distribution on the remaining variables is dimension-independent, with variance 2 from the randomness of $\mathbf{W}$.

Let $\widetilde{\mathbf{M}} = [\mathbf{m}_1, \ldots, \mathbf{m}_q]$ be the observed lower triangular parts of $\Delta\mu$, where $\mathbf{m}_i \in \mathbb{R}^{n-i+1}, \forall i \in [q]$. Let $\mathbf{Q} = [\mathbf{q}_1, \ldots, \mathbf{q}_q]$ where $\mathbf{q}_i \in \mathbb{R}^n, \forall i \in [q]$ be the queries. By **Fact** A.5, for any $\mathbf{g} \in \mathcal{E}$ (an event of probability at least $10\delta$),

$$\mathcal{D}_{KL}(\mathcal{P}'|\mathbf{g} \parallel \mathcal{Q}'|\mathbf{g}) \leq \mathcal{D}_{KL}(L_{\mathcal{P}'}|\mathbf{g} \parallel L_{\mathcal{Q}'}|\mathbf{g})$$

$$\leq \sum_{i=1}^{q} \|C\log^{3/2}(\frac{1}{\delta})\mathbf{m}_i\|_2^2$$

$$\leq C^2 \log^3(\frac{1}{\delta}) \sum_{i=1}^{q} \|\frac{\mathbf{g}\mathbf{g}^T}{\|\mathbf{g}\|_2^2}\mathbf{q}_i\|_2^2$$

$$= C^2 \log^3(\frac{1}{\delta}) \sum_{i=1}^{q} \langle \frac{\mathbf{g}}{\|\mathbf{g}\|_2}, \mathbf{q}_i \rangle^2$$

$$= C^2 \log^3(\frac{1}{\delta}) \frac{\|\mathbf{g}^T\mathbf{Q}\|_2^2}{\|\mathbf{g}\|_2^2}$$

$$\leq \frac{1}{50}$$

By **Fact** A.6, since conditioning (on $\mathbf{g}$) increases KL divergence between $\mathcal{P}'$ and $\mathcal{Q}'$, let $f(\mathbf{g})$ be the conditional probability density of $\mathbf{g}$ on $\mathcal{E}$. Then,

$$\mathcal{D}_{KL}(\mathcal{P}' \parallel \mathcal{Q}') \leq \int_{\mathbf{g}} \mathcal{D}_{KL}(\mathcal{P}'|\mathbf{g} \parallel \mathcal{Q}'|\mathbf{g}) f(\mathbf{g}) d\mathbf{g} \leq \mathcal{D}_{KL}(\mathcal{P}'|\mathbf{g} \parallel \mathcal{Q}'|\mathbf{g}) = \frac{1}{50}$$

By Pinsker's inequality, given $\mathcal{E}$ happens,

$$\mathcal{D}_{TV}(\mathcal{P}' \parallel \mathcal{Q}') \leq \sqrt{\frac{1}{2}\mathcal{D}_{KL}(\mathcal{P}' \parallel \mathcal{Q}')} = \sqrt{\frac{1}{100}} < \frac{1}{3}$$

If the total variation distance between any two distributions $\mathcal{P}'$ and $\mathcal{Q}'$ is at most $\delta$, then any algorithm that distinguishes between $\mathcal{P}'$ and $\mathcal{Q}'$ can succeed with probability at most [2] $\frac{1}{2} + \frac{\delta}{2}$.

---

[2]For two arbitrary distributions $\mathcal{P}'$ and $\mathcal{Q}'$, let the total variation distance between them be $\mathcal{D}_{TV}(\mathcal{P}' \parallel \mathcal{Q}') =$

Since $\mathcal{D}_{TV}(\mathcal{P}' \parallel \mathcal{Q}') \leq \frac{1}{3}$ in our case, this implies that any algorithm for distinguishing $\mathcal{P}'$ and $\mathcal{Q}'$ can succeed with probability at most $\frac{1}{2} + \frac{1}{2} \cdot \frac{1}{3} = \frac{2}{3}$, and so fails with probability $> \frac{1}{3}$. Since $\Pr[\mathcal{E}] \geq 10\delta$, the overall failure probability of an algorithm for distinguishing $\mathcal{P}$ from $\mathcal{Q}$ is thus $10\delta \cdot \frac{1}{3} > \delta$. This implies that to achieve success probability at least $1 - \delta$, $q = \Omega(\frac{\log(1/\delta)}{\log\log(1/\delta)})$.

$\square$

---

$\sup_{\mathcal{E}} |\mathcal{P}'(\mathcal{E}) - \mathcal{Q}'(\mathcal{E})| = \delta$, where $\mathcal{E}$ is an event. Consider an algorithm $\mathcal{A}$ that distinguishes samples from $\mathcal{P}'$ or $\mathcal{Q}'$, and an arbitrary sample $\mathbf{x}$. Let $\mathcal{E} = \Pr[\mathcal{A}(\mathbf{x}) = \mathcal{P}', \mathbf{x} \sim \mathcal{P}']$. If $\mathcal{A}$ succeeds with probability $\geq \frac{1}{2} + \frac{\delta}{2}$, then this implies $\Pr[\mathcal{A}(\mathbf{x}) = \mathcal{P}', \mathbf{x} \sim \mathcal{P}'] \geq \frac{1}{2} + \frac{\delta}{2}$, and $\Pr[\mathcal{A}(\mathbf{x}) = \mathcal{P}', \mathbf{x} \sim \mathcal{Q}'] \geq \frac{1}{2} + \frac{\delta}{2} - \delta = \frac{1}{2} - \frac{\delta}{2}$. This also implies $\Pr[\mathcal{A}(\mathbf{x}) = \mathcal{Q}', \mathbf{x} \sim \mathcal{Q}'] \leq 1 - (\frac{1}{2} - \frac{\delta}{2}) = \frac{1}{2} + \frac{\delta}{2}$, which means the success probability $\mathcal{A}$ is at most $\frac{1}{2} + \frac{\delta}{2}$.