# OpenReview forum: "Optimal Sketching for Trace Estimation"
_NeurIPS.cc/2021/Conference — NeurIPS 2021 Spotlight_

### Official Review · Reviewer_jykV · 2021-07-15

**Rating:** 7
**Confidence:** 4

**Summary:**

The paper studies the problem of estimating the trace of an input PSD matrix with the perspective of improving the query complexity of matrix-vector products required to compute an eps-multiplicative approximation of the trace. There have been two types of randomized algorithms solving this problem: adaptive and non-adaptive.

The former uses samples that depend on previous ones and therefore, by definition, is sequential. The latter doesn't, and is therefore amenable to parallelization, clearly a useful/desirable feature to have.

However, the prior lowest query complexity results for adaptive algorithms were lower than those for non-adaptive algorithms, by a factor of sqrt(log(1/delta)), where delta is the error probability. This paper focuses on non-adaptive algorithms for trace estimation of PSD matrices. Within the framework of this problem, it has two key contributions.

1. The first is an improved analysis of a non-adaptive algorithm proposed in a previous paper. The analysis is an improvement in the sense that it is able to show that this non-adaptive algorithm in fact needs just as many queries as the best adaptive algorithm, thus shaving off a factor of sqrt(log(1/delta)) from the prior analysis.

2. The second contribution is a proof that the new result obtained is in fact optimal, thereby completely resolving the question of non-adaptive query complexity of matrix-vector products for trace estimation of PSD matrices.

In the process of 1., the paper also provides the following technical contribution: a proof that to estimate the k-rank approximation of a PSD matrix with probability 1-delta, it suffices to use O(k + log(1/delta)) queries, down from the previous best query complexity of O(k log (1/delta)). The paper explains how, to achieve k-rank approximation, one needs matrices that satisfy two properties: (1) a subspace embedding property, and (2) a property about orthogonal matrix products. While property (1) was satisfiable by the improved query complexity before, it wasn't known that (2) was. This paper shows, by carefully arguing about tail bounds of chi-squared distributed variables, that (2) is also in fact true for O(k + log(1/delta)) queries.

**Ethical Concerns:**

None, as far as I can tell.

**Ethics Review Area:**

["I don’t know"]

**Limitations And Societal Impact:**

None, as far as I can tell.

**Main Review:**

REVIEW: This is a very strong paper. Both the main technical contributions (listed in the previous field) are strong in their own right: one, giving an improved analysis of an existing algorithm, and second, showing that this is the best possible query complexity for this problem, thereby effectively closing this specific problem. Further, the k-rank approximation using O(k + log(1/delta)) queries is also an improvement of a problem so fundamental, it's taught in week 4 of an introductory machine learning class. Clearly, this paper scores highly on significance and originality.

The paper scores average on clarity: some parts of it are great! I particularly liked the succinct description of the main technical idea in lines 55-60 and of the technical contribution from lines 108 - 111. However,  ​I have some specific parts of the introduction I'd love to see re-written or explained better.

1. The paper's opening paragraph (lines 20-26) gives a laundry list of applications that invoke trace estimation as a subroutine. However,  it's quite difficult to actually read five other papers to see *how* this routine actually shows up there. So, I feel the authors should "open up" at least one (maybe the most impactful) of these applications (in just a couple of sentences) and describe the exact use of trace estimation therein. Though I don't need convincing that trace estimation is important, I am curious to see exactly where it's applicable.

2. Given that the paper measures efficiency in terms of number of queries to matrix-vector products, the authors should explain at the very outset (paragraph 2: lines 27-33) *what* is the exact connection between trace estimation and matrix-vector products.

To  summarize, I enjoyed the technical contributions and think that the problem is well-motivated. However, I think explaining the fundamental assumptions and claimed applications of your work is critical  to making the paper accessible to people who may not have done research on this topic before.

I currently rate this paper 7/10 (GOOD PAPER, ACCEPT). However, I will happily give this  paper a  CLEAR ACCEPT (SCORE 8) , CONDITIONED ON the authors addressing points 1 and 2 above. I look forward to reading the authors' response.


**Time Spent Reviewing:**

4

---

> ### Author Response · Authors · 2021-08-10
> **Response to Reviewer jykV**
>
> We thank the reviewer for the comments.
> We will add the following explanations on estimating the log determinant via trace estimation, with downstream applications to estimating the log probability of Gaussian processes, as well as a description of the importance of the matrix-vector product model:
>
> $\textbf{Gaussian Processes:}$  Gaussian Processes (GP), widely used in Bayesian Optimization, are a popular non-parametric kernel-based method in physics [6], machine learning [7], and other fields such as transportation [3,4,5], environment [1,2], or robotics [8]. During training of GP, the calculation of the marginal log-likelihood contains a heavy-computation
> term, i.e., the log determinant of the covariance matrix, $\log(\det(\mathbf{K}))$, where $\mathbf{K}$ is of size $n $, and $n$ is the number of data points.
> The canonical way of computing $\log(\det(\mathbf{K}))$ is via Cholesky decomposition on $\mathbf{K}$, whose time complexity is $O(n^3)$.
> Since $\log(\det(\mathbf{K})) = \sum_{i=1}^{n}\log(\lambda_i)$, where $\lambda_i$'s are the eigenvalues of $\mathbf{K}$, one can compute $\text{tr}(\log(\mathbf{K}))$ instead. Trace estimation combined with polynomial approximation (e.g. the Chebyshev polynomial, Stochastic Lanczos Quadrature) to $\log$ [20], or trace estimation combined with maximum entropy method [9] provide fast ways of estimating $\text{tr}(\log(\mathbf{K}))$ for large-scale data.
>
> $\textbf{The Importance of Non-adaptivity. }$ We further note the importance of non-adaptive matrix vector queries that both NA-Hutch++ and Hutchinson enjoy. During training GP, the marginal log-likelihood is often computed repeatedly on different covariance matrices $\mathbf{K}$ with updated parameters. Each computation of the log-likelihood requires an estimation of a different $\mathbf{K}$.
> Hutch++ will thus need to compute a small QR decomposition (whose time complexity depends on the number of matrix-vector queries) on each different covariance matrix and the running time can be accumulated across epochs, while NA-Hutch++ and Hutchinson's method, which  only require non-adaptive queries, do not have such a time-consuming step. Furthermore, algorithms with non-adaptive queries like NA-Hutch++ can be easily parallelized across multiple machines, and they correspond to sketching algorithms that are the basis for many streaming algorithms with low memory, or distributed protocols with low-communication overhead.
>
> $\textbf{Matrix-Vector Product Model:}$ In this model, there is an underlying matrix $A$, which is often implicit, and for which one's only access to $A$ is via matrix-vector products. Namely, the algorithm chooses a query vector $v^1$, obtains the product $A \cdot v^1$, chooses the next query vector $v^2$, which is any randomized function of $v^1$ and $A \cdot v^2$, then receives $A \cdot v^2$, and so on. The minimal number $q$ of queries needed by the algorithm to solve a problem with some constant probability is referred to as the $\textit{query complexity}$. In the non-adaptive setting, when $v^1, \ldots, v^q$, are chosen before making any queries to $A$, this is the $\textit{sketching model}$, which is studied on its own (see, e.g., [10]), and in the context of data streams (see, e.g., [11]).
>
> This model and related vector-matrix-vector query models were formalized for a number of problems in [12,13]. For the problem of estimating the top eigenvector, nearly tight bounds were obtained in [14,15]. The motivation for this model is primarily to the setting when $A$ is not represented explicitly.
> For example, one may be given $A = U \Sigma V^\top$ and a function $f:\mathbb{R} \to \mathbb{R}$, and want to compute matrix vector products with the generalized matrix function $f(A) = V f(\Sigma) V^\top$, where $U$ has orthonormal columns, $V^\top$ has orthonormal rows, $\Sigma$ is a diagonal matrix, and $f$ is applied entry-wise to each entry on the diagonal. The covariance matrix corresponds to $f(x) = x^2$, and other common functions $f$ include the matrix exponential $f(x) = e^x$ and other low-degree polynomials $f$, e.g., when $A$ is the adjacency matrix
> of an undirected graph $f(x) = x^3/6$ is used to count the number of triangles. Thus, one is interested in the trace of $f(A)$.
> Another example is when $A$ is the Hessian $H$ of a neural network with a huge number of parameters,
> and it is often impossible to compute or store the entire Hessian [16]. One can compute $H \cdot v$ for any chosen vector $v$ with Pearlmutter's trick [17]. However,  even with Pearlmutter's trick and distributed computation on modern GPUs, it takes 20 hours to compute the eigendensity of a single Hessian $H$ with respect to the cross-entropy loss on the CIFAR-10 dataset [18], from a set of fixed weights for ResNet-18 which has approximately 11 million parameters [16, 19]. This time is directly
> proportional to the number of matrix-vector products, and therefore minimizing this quantity is crucial.
>
> ============
>
> [1] T. J. Ansell et al. Daily mean sea level pressure reconstructions for the European-North Atlantic region for the period 1850-2003. J. Climate, 19(12):2717–2742, 2006.
>
> [2] J. M. Dolan, G. Podnar, S. Stancliff, K. H. Low, A. Elfes, J. Higinbotham, J. C. Hosler, T. A. Moisan, and J. Moisan. Cooperative aquatic sensing using the telesupervised adaptive ocean sensor fleet. In Proc. SPIE Conference on Remote Sensing of the Ocean, Sea Ice, and Large Water Regions, volume 7473, 2009.
>
> [3] J. Chen, K. H. Low, C. K.-Y. Tan, A. Oran, P. Jaillet, J. M. Dolan, and G. S. Sukhatme. Decentralized data fusion and active sensing with mobile sensors for modeling and predicting spatiotemporal traffic phenomena. In Proc. UAI, pages 163–173, 2012.
>
> [4] Jie Chen, Kian Hsiang Low, and Colin Tan. Gaussian process-based decentralized data fusion and active sensing for mobility-on-demand system. Robotics: Science and System, 06 2013.
>
> [5] J. Hensman, N. Fusi, and N. D. Lawrence. Gaussian processes for big data. In Proc. UAI, pages 282–290, 2013.
>
> [6] FRAZIER, Peter I. A tutorial on Bayesian optimization. arXiv preprint arXiv:1807.02811, 2018.
>
> [7] DAMIANOU, Andreas; LAWRENCE, Neil D. Deep gaussian processes. In: Artificial intelligence and statistics. PMLR, 2013. S. 207-215.
>
> [8] N. Xu, K. H. Low, J. Chen, K. K. Lim, and E. B. Özgül. GP-Localize: Persistent mobile robot localization using online sparse Gaussian process observation model. In Proc. AAAI, pages 2585–2592, 2014.
>
> [9] Jack K. Fitzsimons, Diego Granziol, Kurt Cutajar, Michael A. Osborne, Maurizio Filippone, Stephen J. Roberts:
> Entropic Trace Estimates for Log Determinants. ECML/PKDD (1) 2017: 323-338
>
> [10] David P. Woodruff: Sketching as a Tool for Numerical Linear Algebra. Found. Trends Theor. Comput. Sci. 10(1-2): 1-157 (2014)
>
> [11] S. Muthukrishnan: Data Streams: Algorithms and Applications. Found. Trends Theor. Comput. Sci. 1(2) (2005)
>
> [12] Xiaoming Sun, David P. Woodruff, Guang Yang, Jialin Zhang: Querying a Matrix Through Matrix-Vector Products. ICALP 2019: 94:1-94:16
>
> [13] Cyrus Rashtchian, David P. Woodruff, Hanlin Zhu: Vector-Matrix-Vector Queries for Solving Linear Algebra, Statistics, and Graph Problems. APPROX/RANDOM 2020: 26:1-26:20
>
> [14] Max Simchowitz, Ahmed El Alaoui, Benjamin Recht: Tight query complexity lower bounds for PCA via finite sample deformed wigner law. STOC 2018: 1249-1259
>
> [15] Mark Braverman, Elad Hazan, Max Simchowitz, Blake E. Woodworth: The Gradient Complexity of Linear Regression. COLT 2020: 627-647
>
> [16] Behrooz Ghorbani, Shankar Krishnan, Ying Xiao: An Investigation into Neural Net Optimization via Hessian Eigenvalue Density. ICML 2019: 2232-2241
>
> [17] Barak A. Pearlmutter: Fast Exact Multiplication by the Hessian. Neural Comput. 6(1): 147-160 (1994)
>
> [18] Alex Krizhevsky, Geoffrey Hinton: Learning multiple layers of features from tiny images. 2009
>
> [19] Kaiming He, Xiangyu Zhang, Shaoqing Ren, Jian Sun: Deep Residual Learning for Image Recognition. CVPR 2016: 770-778
>
> [20] Kun Dong, David Eriksson, Hannes Nickisch, David Bindel, Andrew Gordon Wilson: Scalable Log Determinants for Gaussian Process Kernel Learning. NeurIPS 2017: 6330–6340.

---

### Official Review · Reviewer_j4ZM · 2021-07-16

**Rating:** 6
**Confidence:** 4

**Summary:**

This paper presents an improved analysis for Hutch++, a stochastic trace estimation method proposed recently. In particular, the paper presents a new analysis for the non-adaptive variant of the Hutch++ algorithm and show that even the non-adaptive algorithm achieves optimal dependency on the failure probability \delta (i.e., O(\sqrt{\log(1/\delta)}))for the number of samples/matvecs. The paper also presents an improved lower bound for the number of matrix-vector queries required in terms of \delta and error tolerance \epsilon. Some numerical results are also presented to illustrate the performance of the methods.

**Limitations And Societal Impact:**

There does not seem to be a discussion on the limitations and potential negative societal impact of their work in the paper.


**Main Review:**

The paper presents a new analysis for the Hutch++ method that improves the non-adaptive query complexity of the algorithm. It also presents an improved lower bound for the number of matrix-vector queries required. These results are certainly interesting from theoretical analysis (TCS) perspective. The numerical results presented and the python implementation with parallel processing will be useful for practitioners.

However, I have the following comments about the paper:

1. Novelty is limited: The main concern about the paper is that the work is very incremental and novelty seems limited. The main algorithms of Hutch++ and much of the analysis have already appeared in reference [16]. The proof of the main theorem 3.1 largely depends on the results in [16] (Although, I have not gone over the proof that is in the supplement in details). The theoretical improvement is really just a \sqrt{\log} factor wrt. failure probability, and does not have any implications on the actual usage and implementation of the algorithm. The lower bound analysis wrt. \delta is novel, but it is not too far from the lower bounds presented in [16] (analysis approach also appears to be similar to [16]).

Therefore, it is not very clear that the publication of this paper in an AI/ML conference is warranted. To my understanding, this paper is  more suitable for Theoretical Computer Science (TCS) or Numerical Linear Algebra (NLA) venue.

2. Minor comment:
I suggest changing the title of the paper since it is almost identical to the Hutch++ paper.

--------------------------
After Author response:

I thank the authors for their response to reviewers' comments and for clarifying things.
One thing I should clarify, which was perhaps confusing, is the comment "much of the analysis has already appeared in [16]", where I was referring to the type of analysis presented, i.e., the multiplicative error bounds and lower bound analysis results that are also developed in [16]. The proof of Theorem 3.1 in this paper does follow the low rank approximation route (similar to [16] and uses the result on splitting the trace estimation), since the algorithm uses sketching to capture the top k singular values (and subspace). This is by design and the method was perhaps designed to leverage the sketching results. To my understanding, the algorithmic advancement was mainly this, using sketching to approximate the top singular values, and then using deflation and applying the stochastic trace estimator on the deflated matrix to capture the remaining part of the spectrum. This key idea has appeared in [16].
However, the novelty in this paper seems to be the new analysis of the sketched low rank approximation (I am not sure if there are changes to the algorithm itself, I request the authors to clarify). This analysis could be of independent interest, which I agree with the authors. Maybe authors can highlight this more and center some of the discussions around this point. Authors can add bullet points in 1.1 to highlight and discuss the key differences between [16] and this paper.

Overall, I agree with other reviewers that the theoretical results are strong. I am happy if the paper is accepted, since the theoretical results might be for interest for other RandNLA problems, including low rank approximation, spectral sums tr(f(A), etc.
But, I am still not convinced entirely that the results are of interest to broad AI readers and practitioners, since the main algorithmic advances have already appeared in [16], and the theoretical improvement of sqrt log factor wrt. to the success probability will likely have no practical consequence when the method is applied to the AI applications listed by the authors. Hence I am increasing the score to 6: Marginally above the acceptance threshold.

**Time Spent Reviewing:**

5

---

> ### Author Response · Authors · 2021-08-10
> **Response to Reviewer j4ZM**
>
> We thank the reviewer for their feedback, though we strongly disagree that much of the analysis has already appeared in [16], and outline the differences in both upper and lower bounds below:
>
> $\textbf{Upper Bound:}$ note that [16] uses an $\textit{off-the-shelf}$ low rank approximation algorithm, whereas we design a new non-adaptive low rank approximation algorithm and analysis in the high probability regime. The new low rank approximation algorithm is in fact the novel idea in our algorithm. Note that low rank approximation is arguably as fundamental as the trace estimation problem itself, and our low rank approximation algorithm can thus be plugged into other applications to improve the dependence on the failure probability $\delta$. Surprisingly, optimal bounds in terms of $\epsilon$ and $\delta$ were not known even for the low rank approximation problem! Our algorithm is non-adaptive and thus applicable in streaming and parallel settings. We note that our reduction of $\log(1/\delta)/\epsilon$ to $\sqrt{\log(1/\delta)}/\epsilon$ is arguably surprising, since it demonstrates that our high probability algorithms provably do better than simply repeating a constant probability algorithm $\log(1/\delta)$ times independently.
>
> ${\bf Lower Bound:}$ our technique is fundamentally different than both the adaptive lower bound in [16], which is based on communication complexity, as well as the non-adaptive lower bound in [16], which is based on distinguishing negatively spiked covariance matrices. Indeed, first, we do not use communication complexity at all. Second, the hard distributions for both of our lower bounds are completely unrelated to those in [16]. As evidence of this, notice that our $\Omega(\sqrt{\log(1/\delta}/\epsilon)$ lower bound based on Wigner matrices, which holds for $\epsilon < 1/\sqrt{\log(1/\delta)}$,
> holds also for adaptive algorithms, and when $\delta$ is constant gives a new proof of the $\Omega(1/\epsilon)$ adaptive lower bound in [16] which does not use communication complexity at all. Second, our additive $\tilde{\Omega}(\log(1/\delta))$ lower bound, which holds for non-adaptive algorithms, uses a new hard distribution. As evidence of this, notice that the hard distribution for non-adaptive algorithms in [16] involves a matrix which is $1/\epsilon \times 1/\epsilon$ and thus can be solved with $\delta = 0$ with only $1/\epsilon$ matrix-vector product queries, by reconstructing the input! It thus has no hope of being used to obtain a dependence on $\delta$ in the lower bound.
>
> Thus, we do not see the overlap in the techniques and kindly ask the reviewer to clarify if we are misunderstanding something. We are happy to explain this better in the paper if that is the issue.
>
> Regarding the comment on scope: given the large number of applications of implicit trace estimation, to, e.g., approximating the histogram of a spectrum in the form of spectral density and eigenvalue counts, approximating matrix and Schatten norms, estimating log-determinants, e.g., for Gaussian processes, estimating the Estrada index, estimating the number of triangles in a graph, and so on, we believe the scope of these results is important to an AI/ML conference.
>
> We are happy to change the title to, e.g., add the words "with High Probability", or to convey a similar meaning.

---

### Official Review · Reviewer_5vUg · 2021-07-16

**Rating:** 7
**Confidence:** 2

**Summary:**

This paper contains two main contributions. First, they provided an improved analysis on the NA-Hutch++ algorithm to bridge the gap between adaptive and non-adaptive algorithm. Secondly, they also provided a nearly matching lower-bound on the complexity of the problem, closing the gap.

**Limitations And Societal Impact:**

Based on my limited knowledge, I believe that it is meaningful in some applications to break down the $1\pm\epsilon$-multiplicative bound into an upper bound and lower bound. I am wondering if some of the results can be refined for these error bound or whether they are nearly the same.

**Main Review:**

I acknowledge that I am not an expert on this particular problem.Closing the complexity gap is usually a significant contribution to any problem, which indicates that improvement should be sought elsewhere. Therefore, I believe the authors have made significant contribution to the trace estimation problem where we only have access to a linear projection oracle.

The paper is well-written. Even if I did not read the proof details, the authors have kindly provided intuitions and key idea behind the proof.

My only concern is that the problem and the application itself might still not be close enough to make direct impact on real world applications, but the theoretical advances should interest the audience.

update: thanks for the comment

**Time Spent Reviewing:**

1

---

> ### Author Response · Authors · 2021-08-10
> **Response to Reviewer 5vUg**
>
> We thank the reviewer for their comments.
>
> We refer the reader to Ubaru and Saad's survey "Applications of Trace Estimation Techniques" from HPCSE, 2017, for a discussion of real-world applications of trace estimation, including approximating the histogram of a spectrum in the form of spectral density or eigenvalue counts or numerical rank, to approximating matrix and Schatten norms, to estimating log-determinants, e.g., for Gaussian processes, to estimating the Estrada index, to estimating the number of triangles in a graph, and to clustering and the problem of community detection in social graphs. Non-adaptive algorithms allow for increased parallelism, as is often necessary for large-scale data sets.
>
> In addition to improved applications of trace estimation, we give the first $O(1)$-approximate low rank approximation with $O(k + \log (1/\delta))$ non-adaptive matrix-vector products. Since non-adaptive algorithms can be implemented in a data stream, we directly improve the state of the art streaming algorithm for principal component analysis (PCA) of Boutsidis, Woodruff, and Zhong "Optimal Principal Component Analysis in Distributed and Streaming Models" from STOC, 2016, improving the space complexity of their streaming algorithm when $\epsilon = \Theta(1)$ from $O((k \log(1/\delta))\cdot m)$ words to $O((k + \log(1/\delta)) \cdot m)$ words, where m is the dimension of the feature vector. Computing a low rank approximation of streaming data with low memory is an important practical problem for providing dimensionality reduction in online learning with limited memory; please see the line of previous work obtaining a series of improved bounds in the abovementioned paper.

---

### Official Review · Reviewer_H6Dn · 2021-07-18

**Rating:** 8
**Confidence:** 3

**Summary:**

This paper gives improved bounds for estimating the trace of a matrix using matrix-vector products. It turns what was previous a multiplicative factor of log(1 / failure probability) into an additive one, and shows that this bound is tight up to a loglog factor. These methods are evaluated on a wide range of matrices, several of which are implicitly represented (via the Lanczos method), and a better success probability was observed.

**Limitations And Societal Impact:**

My impression is that failure probability are usually an artifact of the analysis. Outer loops that utilize these trace estimators are often ok with some randomized noise in the answer, and I feel empirical results (in either positive or negative directions) on this would be interesting.

Also, the performance gains of these methods shown in Fig 1 seems to be around 1.5x, which to me feels rather marginal. It might take more end-to-end experiments to demonstrate more pronounced gains of these methods. I'm very happy that the authors took time to provide additional empirical evaluations in their responses, and am now quite interested in running my own experiments on this topic.

**Main Review:**

I believe this is a solid result: trace estimators have been well studied, and the dependence on success probability is a nature parameter for analyzing algorithmic performances. The paper essentially closes off this dependence issue by showing that a additive log(1 / failure probability) is sufficient, and also necessary up to a loglog factor. They also empirically demonstrate this improvement by showing the failure probability of this method versus previous, and raises some interesting questions about better parallel performances of such estimators.

The importance of trace estimators across numerical/ML algorithms make this paper a clear accept. On the other hand, I feel the experiments demonstrate these gains are rather limited, and occur in performance only when parallelism is brought in. However, I believe they provide natural directions to extend some of the primitives developed here, and the evidences provided in the responses indicate that significantly more can be done on this front.

**Time Spent Reviewing:**

2

---

> ### Author Response · Authors · 2021-08-10
> **Response to Reviewer H6Dn**
>
> We thank the reviewer for their comments.
>
> We give an additional experiment here to illustrate the benefits of our method, i.e., to show the performance of Hutch++ and NA-Hutch++ on log determinant estimation of a covariance matrix, i.e., for computing $\log(\det(\mathbf{K})) = \text{tr}(\log(\mathbf{K}))$. Computing $\log(\det(\mathbf{K}))$ is required when computing the marginal log-likelihood in Gaussian Process models and is often a bottleneck in the running time on large scale data. Please see our response to Reviewer \#jykV for more background on Gaussian Processes; here we focus on empirical results.
>
> Recently, [9] proposed an improved method for log determinant estimation. This method first uses Hutchinson's trace estimation as a subroutine to estimate up to the $k$-th moments of the eigenvalues, given a fixed $k$. The $i$-th moment of the eigenvalues is $\mathbb{E}[\lambda^{i}] = \frac{1}{n}\text{tr}(\mathbf{K}^{i})$, where $\mathbf{K}$ is an $n \times n$ PSD matrix, and $\lambda$ is the vector of eigenvalues. The estimated moments are then used as a constraint in maximum entropy estimation that estimates the probability density of the  eigenvalues. This distribution is later used to compute an estimate of the log determinant. [9] shows that their proposed approach outperforms traditional Chebyshev/Lanczos methods for computing $\log(\det(\mathbf{K}))$ in terms of absolute value of the relative error, i.e. abs(estimated log determinant - true log determinant)/abs(log determinant).
>
> We compare the estimated log determinant of a covariance matrix with different subroutines for estimating the moments of the  eigenvalues: Hutchinson's method as in [9], Hutch++, and NA-Hutch++. We use 4 PSD matrices from the standard UFL Sparse Matrix Collection: bcsstk20 (size $485 \times 485$), bcsstm08 (size $1074 \times 1074$), sts4098 (size $4098 \times 4098$) and bcsstm25 (size $15439\times 15439$), with varying max moments $\{10, 15, \dots, 30\}$. We use $30$ matrix-vector queries for trace estimation on the three smaller datasets: bcsstk20, bcsstm08 and sts4098, and $60$ matrix-vector queries for trace estimation on the larger dataset bcsstm25. We repeated each run 100 times and reported the mean estimated log determinant with each trace estimation subroutine.
> Note that Hutchinson needs $O(\log(1/\delta)/\epsilon^2)$ matrix-vector queries to achieve $(1\pm \epsilon)$ approximation to the trace with probability $1-\delta$, while Hutch++ and NA-Hutch++ both need only $O(\sqrt{\log(1/\delta)}/\epsilon + \log(1/\delta))$ matrix-vector queries.
> This implies in theory that with the same number of matrix-vector queries, Hutch++/NA-Hutch++ will give more accurate estimation of the eigenvalue moments, and hopefully a more accurate estimation of the log determinant.
> While an improved estimate of the eigenvalue moments does not necessarily lead to an improved estimate of the log determinant, it is not hard to show that an accurate moment estimation does lead to improved log determinant estimation in cases where the eigenspectrum of $\mathbf{K}$ contains a few very large eigenvalues. Such a case will cause Hutchinson's method to have very large variance, while our method reduces the variance by first removing the large eigenvalues.
>
> We show our results in terms of the absolute value of the relative error, following [9], on the above 4 datasets in the following tables:
>
> $\texttt{bcsstk20}$
>
> |Moment | Hutchinson | Hutch ++ | NA-Hutch++|
>
> 10 | 0.1199 | 0.1042 | 0.1068 |
>
> 15 | 0.1157 | 0.1036 | 0.1054 |
>
> 20 | 0.1102 | 0.1016 | 0.1024 |
>
> 25 | 0.1054 | 0.0993 | 0.0995 |
>
> 30 | 0.1018 | 0.0974 | 0.0972 |
>
> $\texttt{bcsstm08}$
>
> |Moment | Hutchinson | Hutch ++ | NA-Hutch++|
>
> 10 | 1.2040 | 1.1871 | 1.1872 |
>
> 15 | 1.1961 | 1.1811 | 1.1812 |
>
> 20 | 1.1878 | 1.1783 | 1.1782 |
>
> 25 | 1.1806 | 1.1743 | 1.1742 |
>
> 30 | 1.1748 | 1.1702 | 1.1701 |
>
> $\texttt{bcsstm25}$
>
> |Moment | Hutchinson | Hutch ++ | NA-Hutch++|
>
> 10 | 0.2684 | 0.2647 | 0.2656 |
>
> 15 | 0.2380 | 0.2350 | 0.2331 |
>
> 20 | 0.1959 | 0.1934 | 0.1899 |
>
> 25 | 0.1744 | 0.1721 | 0.1698 |
>
> 30 | 0.1603 | 0.1583 | 0.1560 |
>
> $\texttt{sts4098}$
>
> |Moment | Hutchinson | Hutch ++ | NA-Hutch++|
>
> 10 | 0.7160 | 0.6951 | 0.6951 |
>
> 15 | 0.7124 | 0.6944 | 0.6944 |
>
> 20 | 0.7085 | 0.6926 | 0.6926 |
>
> 25 | 0.7050 | 0.6907 | 0.6907 |
>
> 30 | 0.7020 | 0.6889 | 0.6889 |
>
>
> From the above tables, one can observe that log determinant estimation with Hutch++ and NA-Hutch++ as the subroutines gives lower absolute value of the relative errors to the estimated log determinant, as compared to Hutchinson's method.
>
> $\textbf{The Importance of Non-adaptivity. }$ We further note the importance of non-adaptive matrix vector queries that both NA-Hutch++ and Hutchinson enjoy. During training of Gaussian Processes, the marginal log-likelihood is often computed repeatedly on different covariance matrices $\mathbf{K}$ with updated parameters. Each computation of log-likelihood requires an estimation of a different $\mathbf{K}$.
> Hutch++ will thus need to compute a small QR decomposition (whose time complexity depends on the number of matrix-vector queries) on each different covariance matrix and the running time can be accumulated across epochs, while NA-Hutch++ and Hutchinson's method, which  only require non-adaptive queries, do not have such a time-consuming step. Furthermore, algorithms with non-adaptive queries like NA-Hutch++ can be easily parallelized across multiple machines, and they correspond to sketching algorithms that are the basis for many streaming algorithms with low memory, or distributed protocols with low-communication overhead.
>
> =============
>
> [9] Jack K. Fitzsimons, Diego Granziol, Kurt Cutajar, Michael A. Osborne, Maurizio Filippone, Stephen J. Roberts:
> Entropic Trace Estimates for Log Determinants. ECML/PKDD (1) 2017: 323-338

---

### Decision · Program_Chairs · 2021-09-27

**Decision:**

Accept (Spotlight)

**Comment:**

This paper considers the fundamental problem of estimating the trace of a PSD matrix from matrix vector product queries. Prior results had a gap between the adaptive and non-adaptive versions of the problem. In this paper, the authors propose a non-adaptive algorithm with nearly the same performance as the best adaptive algorithm. The paper also makes nice contributions in establishing lower bounds on the number of queries. The reviewers all felt that this paper should be published in NeuRIPS.